# Extracellular vesicles from CLEC2-activated platelets enhance dengue virus-induced lethality via CLEC5A/TLR2

Pei-Shan Sung [1], Tur-Fu Huang[2,3] & Shie-Liang Hsieh [1,4,5]

Platelet-leukocyte interactions amplify inflammatory reactions, but the underlying mechanism is still unclear. CLEC5A and CLEC2 are spleen tyrosine kinase (Syk)-coupled C-type lectin receptors, abundantly expressed by leukocytes and platelets, respectively. Whereas CLEC5A is a pattern recognition receptor (PRR) to flaviviruses and bacteria, CLEC2 is the receptor for platelet-activating snake venom aggretin. Here we show that dengue virus (DV) activates platelets via CLEC2 to release extracellular vesicles (EVs), including exosomes (EXOs) and microvesicles (MVs). DV-induced EXOs (DV-EXOs) and MVs (DV-MVs) further activate CLEC5A and TLR2 on neutrophils and macrophages, thereby induce neutrophil extracellular trap (NET) formation and proinflammatory cytokine release. Compared to $stat1^{-/-}$ mice, simultaneous blockade of CLEC5A and TLR2 effectively attenuates DV-induced inflammatory response and increases survival rate from 30 to 90%. The identification of critical roles of CLEC2 and CLEC5A/TLR2 in platelet-leukocyte interactions will support the development of novel strategies to treat acute viral infection in the future.

[1] Institute of Clinical Medicine, National Yang-Ming University, 11221 Taipei, Taiwan. [2] Department of Medicine, Mackay Medical College, 25245 New Taipei City, Taiwan. [3] Department of Pharmacology, College of Medicine, National Taiwan University, 10051 Taipei, Taiwan. [4] Genomics Research Center, Academia Sinica, Taipei 11529, Taiwan. [5] Department of Medical Research, Taipei Veterans General Hospital, 11217 Taipei, Taiwan. Correspondence and requests for materials should be addressed to S.-L.H. (email: slhsieh@gate.sinica.edu.tw)

Acute viral infections frequently cause severe inflammatory reactions such as cytokine storms via the activation of platelets and leukocytes. While platelet–leukocyte interactions are key to the regulation of inflammation they also contribute to the pathogenesis of infectious diseases and play a critical role in vascular injury, thrombosis, and autoimmunity[1–3]. Dengue virus (DV) is the most common arbovirus, with ~2.5 billion people worldwide being at risk of infection[4,5]. DV infection has been shown to activate macrophages and platelets to secrete proinflammatory cytokines and extracellular vesicles (EVs)[6–9]; the most severe responses to DV are systemic inflammation and increased vascular permeability. Here, we investigated how DV activates platelets and leukocytes, and whether DV-induced leukocyte–platelet interactions contribute to disease severity.

Previous studies demonstrated that CLEC5A is the myeloid PRR for DV, Japanese encephalitis virus (JEV)[10] and type A influenza viruses (IAVs)[11]. Moreover, activation of CLEC5A triggers NALP3 inflammasome activation and proinflammatory cytokine release[7,8,12], thereby inducing increased systemic vascular permeability and hemorrhagic shock[8,13]. However, blockade of CLEC5A only provides partial protection in mice challenged with lethal doses of DV[13] and H5N1 IAV[11]. These observations suggest that innate immune receptor(s) other than CLEC5A is (are) involved in DV and H5N1 IAV-induced inflammatory reactions. Recently, we demonstrated that CLEC5A recognizes GlcNAc-MurNAc disaccharides of bacterial peptidoglycans, and is co-activated with toll-like receptor 2 (TLR2) by *Listeria monocytogenes* and *Staphylococcus aureus*, thereby inducing NET formation and proinflammatory cytokine release[14]. This observation indicates that CLEC5A works closely with TLR2 in pathogen-induced inflammatory reactions.

Similar to other myeloid cells, platelets express abundant innate immune receptors, including TLRs, lectins, collagen receptors, and integrins, allowing them to sense various pathogens and initiate inflammatory reactions[15,16]. Previous studies showed that DV activates platelets to release IL-1β-containing microparticles (also denoted as EVs) and increases vascular permeability[9]. EVs, including MVs and EXOs, are membranous vesicles secreted from platelets, hepatocytes, and other cell types[17]; they transfer proteins, microRNAs, peptides, and nucleic acids to recipient cells via membrane fusion to modulate cell functions. In addition, EVs bind to integrins, proteoglycans, lipid-binding proteins, and the phosphatidylserine receptor TIM4, to initiate intracellular signals[17]. Since platelets contribute to NET formation[18], as well as producing EVs, after DV stimulation[9] we were interested to identify the PRR responsible for DV-induced platelet activation.

Here we report that DV activates CLEC5A on neutrophils to induce NET formation. We also show that DV activates platelets via CLEC2, causing them to release EVs that further enhance NET formation and proinflammatory cytokine production through activation of CLEC5A and TLR2 on macrophages and neutrophils. Simultaneous blockade of CLEC5A and TLR2 not only inhibits NET formation and attenuates inflammation, but also dramatically reduces both DV-induced systemic vascular permeability and lethality. This study reveals the critical roles of CLEC2 and CLEC5A/TLR2 in platelet–leukocyte interactions, thereby supporting the development of novel strategies to reduce tissue damage and increase survival of patients with severe DV infection.

## Results

### DV activates platelets via CLEC2 to enhance NET formation.
To determine whether CLEC2 is responsible for DV-induced platelet activation, we incubated DV (PL046 strain) with human platelets in the presence of an antagonistic anti-CLEC2 mAb (Fig. 1a). The expression of platelet activation markers (CD62p and CD63) and EVs numbers were both upregulated by DV; this phenomenon was suppressed by the anti-CLEC2 mAb (Fig. 1a). Moreover, upregulation of CD62p was abolished in mouse CLEC2$^{-/-}$ platelets stimulated with DV (NGC-N strain) and aggretin (CLEC2 ligand) (Fig. 1b). This observation demonstrates the critical role of CLEC2 in DV-induced platelet activation. It has been reported that activated platelets interact with neutrophils to amplify inflammatory reactions and NET formation[1,18], thus we asked whether DV-activated platelets (DV-PLTs) contribute to NET formation. While DV alone was able to induce mild NET formation, this response was enhanced dramatically in the presence of platelets (Fig. 1c). Moreover, DV-PLTs-induced NET formation was inhibited by anti-CLEC2 mAb (Fig. 1d). Furthermore, CLEC2$^{-/-}$ platelets were unable to enhance NET formation (Fig. 1e), even when stimulated with DV or aggretin. These observations suggest that DV stimulation of CLEC2 on platelets enhances NET formation. The structure of NETs formed in response to DV and the expression of CLEC5A and CLEC2 by platelets and neutrophils are illustrated in Supplementary Fig. 1.

### CLEC5A and TLR2 are responsible for DV-PLTs-induced NET formation.
Since CLEC5A has been implicated in bacteria-induced NET formation[14], we asked whether CLEC5A also contributes to DV-induced and DV-PLTs-induced NET formation. Neutrophils were incubated with DV in the presence of an anti-CLEC5A mAb for 3 h; confocal microscopy revealed that DV-induced NET formation was suppressed by anti-CLEC5A mAb in a dose-dependent manner, and was almost completely abolished at the highest concentration (300 μg/ml) of anti-CLEC5A mAb tested (Fig. 2a, left). However, DV-PLTs-induced NET formation was only partially inhibited by anti-CLEC5A mAb under the same conditions (Fig. 2a, right). This observation suggests that DV-PLTs trigger NET formation via both CLEC5A-dependent and CLEC5A-independent pathways. Previously, we demonstrated that bacteria co-activated CLEC5A and TLR2 to enhance NET formation[14], thus we asked whether both receptors are involved in DV-PLTs-induced NET formation. While DV-PLTs-induced and aggretin-PLTs-induced NETs were partially inhibited by either anti-CLEC5A or anti-TLR2 mAbs, simultaneous blockade of both receptors almost completely suppressed NET formation (Fig. 2b). To further confirm the role of CLEC5A and TLR2 in NET formation, we incubated WT mouse platelets with mouse-adapted DV (NGC-N strain) in vitro for 3 h (MOI = 3), followed by addition of WT, *clec5a$^{-/-}$tlr2$^{+/+}$*, *clec5a$^{+/+}$tlr2$^{-/-}$*, and *clec5a$^{-/-}$tlr2$^{-/-}$* mouse neutrophils and observation of NET formation (Fig. 2c). While DV induced NET formation in WT and *clec5a$^{+/+}$tlr2$^{-/-}$* neutrophils, NET formation was not observed in *clec5a$^{-/-}$tlr2$^{+/+}$* and *clec5a$^{-/-}$tlr2$^{-/-}$* neutrophils. In contrast, aggretin did not induce NET formation in any of the mouse neutrophils tested (Fig. 2c). In the presence of WT platelets, both DV-PLTs-induced and aggretin-PLTs-induced NET formation were partially inhibited in *clec5a$^{-/-}$tlr2$^{+/+}$* and *clec5a$^{+/+}$tlr2$^{-/-}$* neutrophils, and were almost completely abolished in *clec5a$^{-/-}$tlr2$^{-/-}$* neutrophils (Fig. 2c). Thus, we conclude that both DV-PLTs and aggretin-PLTs co-activate CLEC5A and TLR2 to induce NET formation.

### DV-induced EVs contribute to platelet-induced NET formation.
Because EVs released from DV-activated platelets (DV-EVs) play critical roles in homeostasis and disease[19,20], we asked whether DV-EVs contribute to platelet-mediated enhancement of

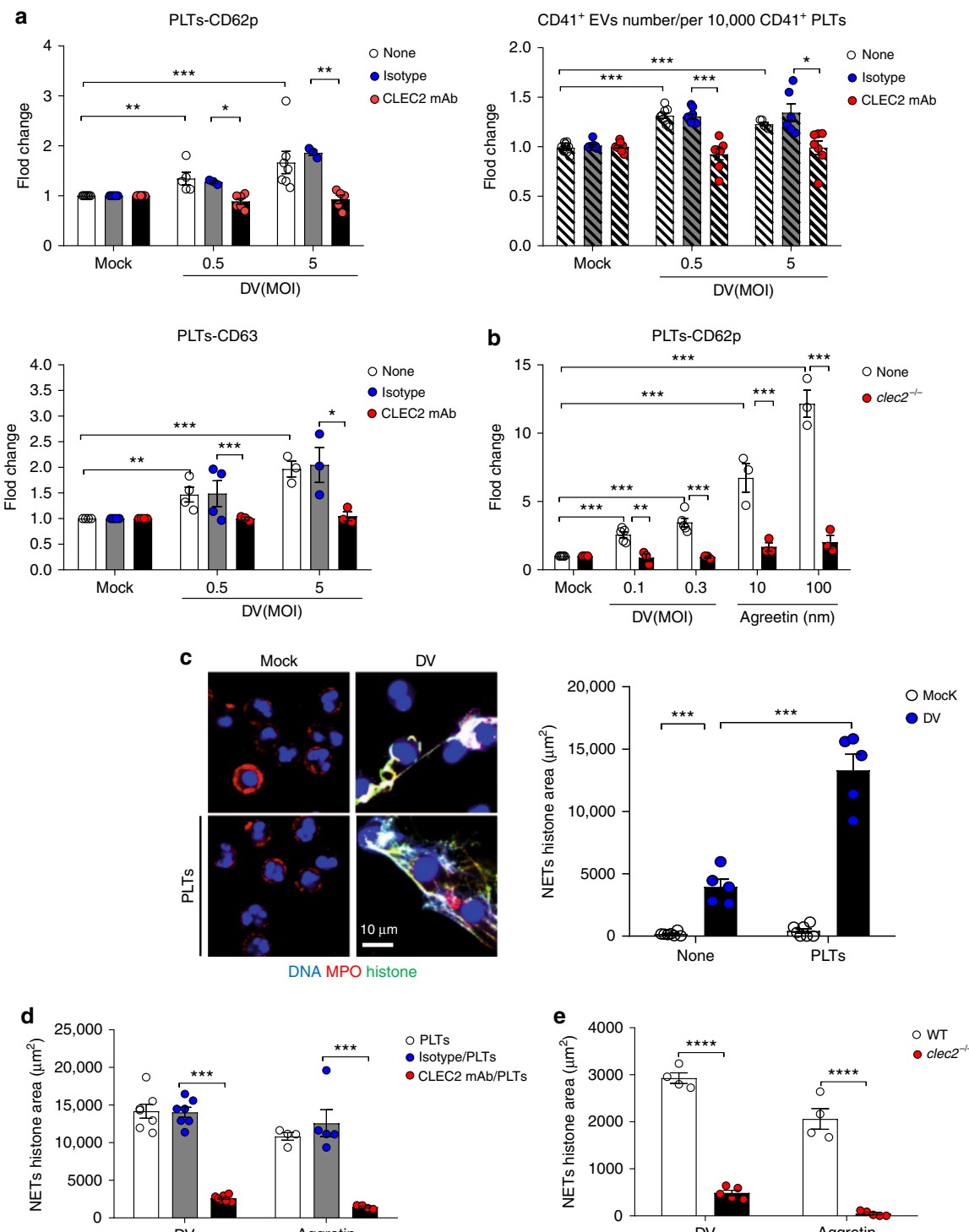

**Fig. 1** DV activates platelets via CLEC2 to enhance NET formation. **a** Human platelets were pre-incubated with anti-CLEC2 mAb or isotype control mAb at room temperature for 15 min, followed by incubation with DV (PL046) at 37 °C for 1 h before subjected to flow cytometry analysis. **b** Platelets from WT and *clec2⁻/⁻* mice were incubated with DV (NCG-N strain) or aggretin for 1 h at 37 °C, followed by flow cytometry analysis. Data are presented as fold change in MFI over mock control. **c** Neutrophils were co-incubated with autologous platelets and DV for 3 h at 37 °C. DNA, histone, and MPO were detected by Hoechst 33342 (blue), anti-histone antibody (green), and anti-MPO antibody (red), respectively. **d** Human platelets were preincubated with isotype control or anti-CLEC2 mAb for 15 min before DV stimulation. **e** Mouse neutrophils were incubated with WT or *clec2⁻/⁻* platelets and stimulated with DV (NGC-N) or aggretin at 37 °C for 3 h. NET area was measured by immunofluorescence staining to determine the histone area (μm²). Data are mean ± SEM from at least three independent experiments. **p < 0.01, ***p < 0.001 (Student's t-test). Scale bar: 10 μm. Source data are provided as Source Data file

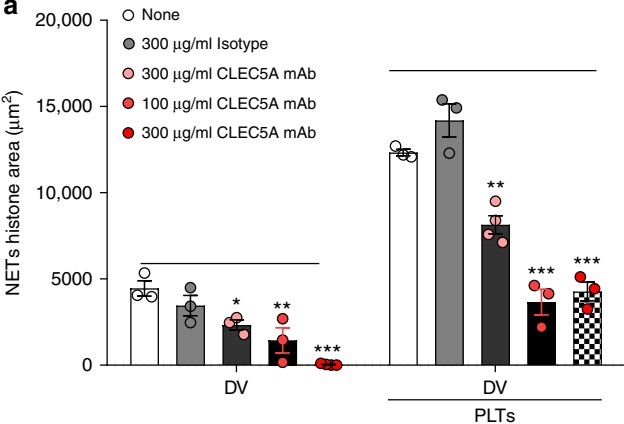

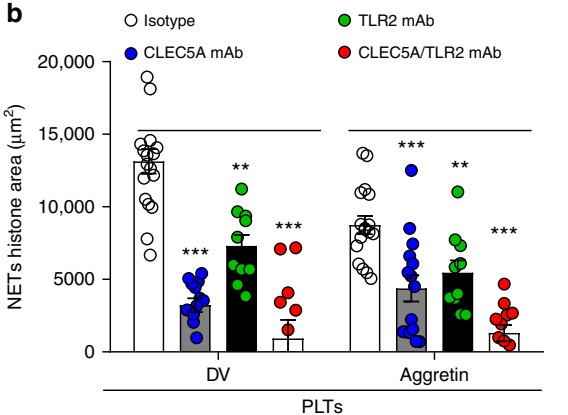

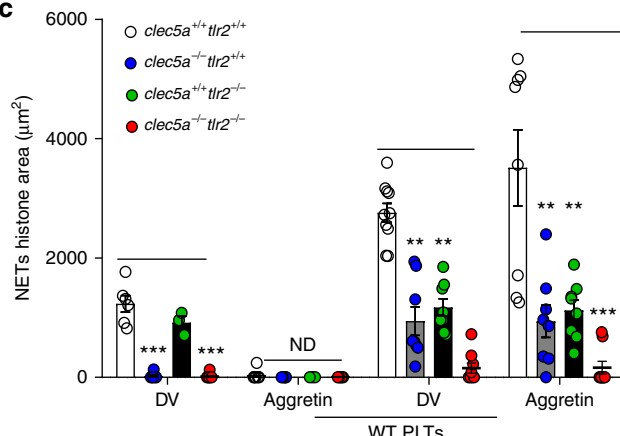

**Fig. 2** CLEC5A and TLR2 mediate DV-activated and DV-activated platelet-induced NET formation. **a** Human neutrophils were pre-treated with isotype control (300 μg/ml) or anti-CLEC5A mAb, followed by incubation with platelets and DV for 3 h at 37 °C. **b** Alternatively, neutrophils were pre-incubated with anti-CLEC5A mAb and anti-TLR2 mAb, followed by incubation with DV-activated platelets (MOI = 5) or aggretin-activated platelets at 37 °C for 3 h. **c** Neutrophils from WT (*clec5a*[+/+]*tlr2*[+/+]), *clec5a*[−/−]*tlr2*[+/+], *clec5a*[+/+]*tlr2*[−/−], and *clec5a*[−/−]*tlr2*[−/−] mice were incubated with WT platelets in the presence of DV (NGC-N) for 3 h at 37 °C. NETs were measured by immunofluorescence staining (μm²). Data are mean ± SEM from at least three independent experiments. *$p < 0.05$, **$p < 0.01$, ***$p < 0.001$ (Student's *t*-test). Source data are provided as Source Data file

**NET formation.** Platelets were pre-incubated with anti-CLEC2 mAb for 15 min after which DV-PLTs-induced expression of CD62p and CD63 was determined in CD41[+] EVs by flow cytometry. While DV-PLTs upregulated the expression of CD62p and

CD63 in CD41[+] EVs, anti-CLEC2 mAb effectively suppressed this (Fig. 3a). We further compared the abilities of DV-PLTs and DV-EVs to induce NET formation (Fig. 3b). We found that DV-EVs and aggretin-EVs induced NET formation with similar potencies to DV-PLTs and aggretin-PLTs (Fig. 3b). Furthermore, EVs from anti-CLEC2 mAb-pretreated platelets had significantly reduced capacity to induce NET formation, suggesting that activation of CLEC2 is required to produce EVs with this activity (Fig. 3c). Similar to DV-PLTs (Fig. 2b), DV-EVs-induced NET formation was almost completely abolished by anti-CLEC5A and anti-TLR2 mAbs (Fig. 3d). Thus, we conclude that platelet-mediated enhancement of NET formation occurs via DV-EVs released from CLEC2-activated platelets.

**Differential effects of microvesicles and exosomes on NET formation.** We purified DV-EVs by centrifugation for nanoparticle tracking analysis (NTA). We found that the numbers of EXOs (50–100 nm) and MVs (>100 nm) from activated platelets were increased after incubation with aggretin and DV, respectively (Fig. 4a). EVs from DV-activated and aggretin-activated platelets mainly fell in the size range 50–150 nm, which is similar to the size distribution of EVs from thrombin-activated platelets (Fig. 4b). In addition, similar expression levels of CD41, HSP70, CD63, CD9, and CD81 were determined in activated platelets and EVs (Fig. 4c). Interestingly, DV-EXOs-induced and aggretin-EXOs-induced NET formation were inhibited efficiently by anti-CLEC5A mAb (Fig. 4d), while DV-MVs-induced and aggretin-MVs-induced NET formation were preferentially inhibited by anti-TLR2 mAb (Fig. 4e). Thus, we conclude that EXOs and MVs from CLEC2-activated platelets preferentially activate CLEC5A and TLR2, respectively, to induce NET formation.

**DV-induced NETs contribute to increased vascular permeability.** We further asked whether DV-induced NETs contribute to increased vascular permeability. As DV-PLTs tended to adhere to endothelial cells, DV-EVs were used in their place for in vitro permeability assays. Whilst large increases in endothelial monolayer permeability were seen in the presence of neutrophils (black bar, Fig. 5a), DNase I reduced this to the same level as DV alone (black bar, Fig. 5a). Furthermore, whereas EVs from non-activated platelets had no effect on permeability (open slashed bar, Fig. 5a), DV-EVs substantially increased permeability and this effect was further enhanced in the presence of neutrophils (black slashed bar, Fig. 5a). These observations suggest that both DV-induced and DV-EVs-induced permeability changes are enhanced by NETs. To understand the roles of CLEC5A and TLR2 in DV-EVs-induced permeability changes, HMEC-1 endothelial cells were incubated with DV-EVs and neutrophils, in the presence or absence of anti-CLEC5A mAb and/or anti-TLR2-blocking antibodies (Fig. 5b). We found that blockade of both CLEC5A and TLR2 inhibited changes in the permeability of endothelial monolayers more effectively than blockade of either CLEC5A or TLR2 alone. We further examined the role of DV-induced NETs in enhancing vascular permeability in vivo using the Evans Blue assay. We observed that permeability changes in vital organs (spleen>small intestine>kidney>liver) after DV infection were inhibited efficiently by DNase I (Fig. 5c). Furthermore, anti-TLR2 mAb was able to further reduce permeability changes in *stat1*[−/−] and *stat1*[−/−]*clec5a*[−/−] mice (Fig. 5d). These observations suggest that DV-induced NETs contribute significantly to increases in systemic vascular permeability, and that blockade of CLEC5A and TLR2 is able to reduce NET formation and permeability changes in vivo.

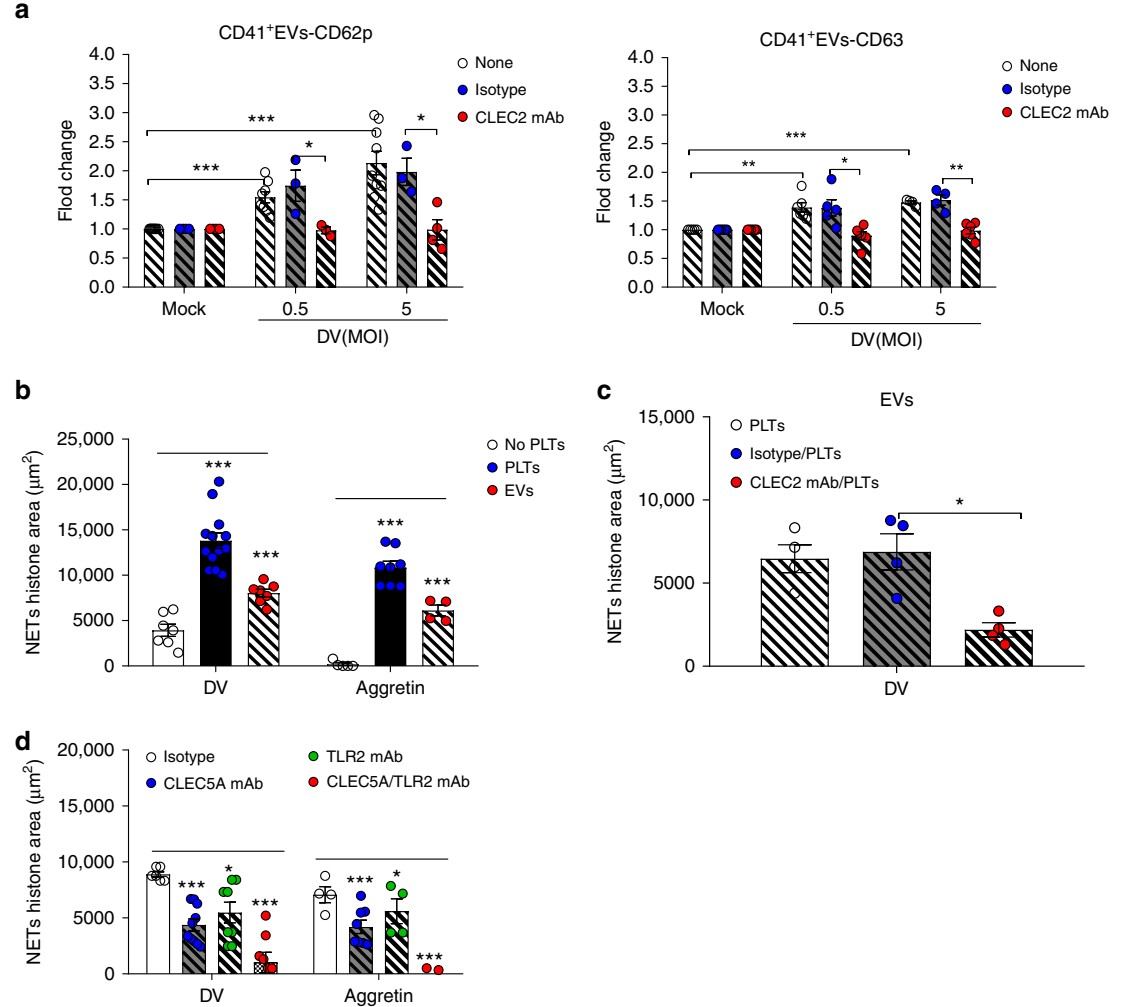

**Fig. 3** DV stimulates platelets to release EVs that induce NET formation via CLEC2. **a** Human platelets were incubated with DV (PL046) at 37 °C for 1 h in the presence or absence of anti-CLEC2 mAb, followed by harvesting of EVs by centrifugation for flow cytometry analysis. **b** Human platelets were incubated with DV (MOI = 0.5) or aggretin (3 nM) at 37 °C for 1 h, followed by harvesting EVs by ultracentrifugation. Both activated platelets and PLT-EVs were incubated with neutrophils and NET formation was observed. **c** Human platelets ($8 \times 10^6$) were preincubated with isotype control or anti-CLEC2 mAb for 15 min before DV (MOI = 0.5) stimulation, followed by harvesting EVs to stimulate human neutrophils and observation of NET formation. **d** Neutrophils ($8 \times 10^5$) were pre-pretreated with isotype control, anti-CLEC5A mAb, or anti-TLR2 mAb at 37 °C for 1 h, followed by incubation with EVs isolated from DV-PLTs ($8 \times 10^6$) and aggretin-PLTs at 37 °C for 3 h. NETs were measured by immunofluorescence staining ($\mu m^2$). Data are mean ± SEM from at least three independent experiments. *$p < 0.05$, **$p < 0.01$, ***$p < 0.001$ (Student's $t$-test). Source data are provided as Source Data file

**DV-induced NET formation occurs via CLEC5A and TLR2 in vivo**. We further examined NET formation after DV infection in vivo, where NETs were characterized by the presence of DNA (Hoechst 33342, blue), MPO (red), and histone (green) in mouse spleens (Supplementary Fig. 2) and quantified by determining the amounts of citrullinated histone H3 (Fig. 6a, c) and MPO (Fig. 6b, c) in spleens. We found that both citrullinated histone H3 and MPO were highly elevated in DV-infected $stat1^{-/-}$ mouse spleens (upper panels, Fig. 6a, b), but significantly reduced in $stat1^{-/-}clec5a^{-/-}$ spleens (lower panels, Fig. 6a, b). Injection of DNase I or anti-TLR2 mAb further reduced the amounts of citrullinated histone and MPO (middle and right columns, Fig. 6a, b). Thus, we conclude that DV-induced NET formation in vivo occurs via activation of CLEC5A and TLR2.

**Simultaneous blockade of CLEC5A and TLR2 inhibits inflammation and increases survival of DV-infected mice**. Proinflammatory cytokines play crucial roles in the pathogenesis of dengue hemorrhagic shock[13], thus we asked whether DV-EVs also trigger proinflammatory cytokine release from macrophages. Compared to DV, DV-EVs induced higher amounts of TNF-α and IL-6 from macrophages (Fig. 7a). Moreover, both anti-CLEC5A mAb and anti-TLR2 mAb inhibited DV-EVs-induced proinflammatory cytokine release in vitro (Fig. 7b). We went on to examine the expression of proinflammatory cytokines in vivo. Compared to $stat1^{-/-}$ mice, $stat1^{-/-}clec5a^{-/-}$ mice produced less proinflammatory cytokines after DV infection (Fig. 7c). Injection of anti-TLR2 mAb further suppressed proinflammatory cytokine production in $stat1^{-/-}$ and $stat1^{-/-}clec5a^{-/-}$ mice (black columns, Fig. 7c). This observation suggests that DV infection leads to activation of CLEC5A-mediated and TLR2-mediated pathways to induce proinflammatory cytokine release.

We investigated the protective effects of anti-TLR2 mAb in $stat1^{-/-}$ and $stat1^{-/-}clec5a^{-/-}$ mice after DV infection. While all $stat1^{-/-}$ mice died by 16 days post-DV infection, $stat1^{-/-}clec5a^{-/-}$ mice were more resistant to DV infection (30% survival at 21 days). Anti-TLR2 mAb delayed, but did not prevent, the death of $stat1^{-/-}$ mice after DV infection (left,

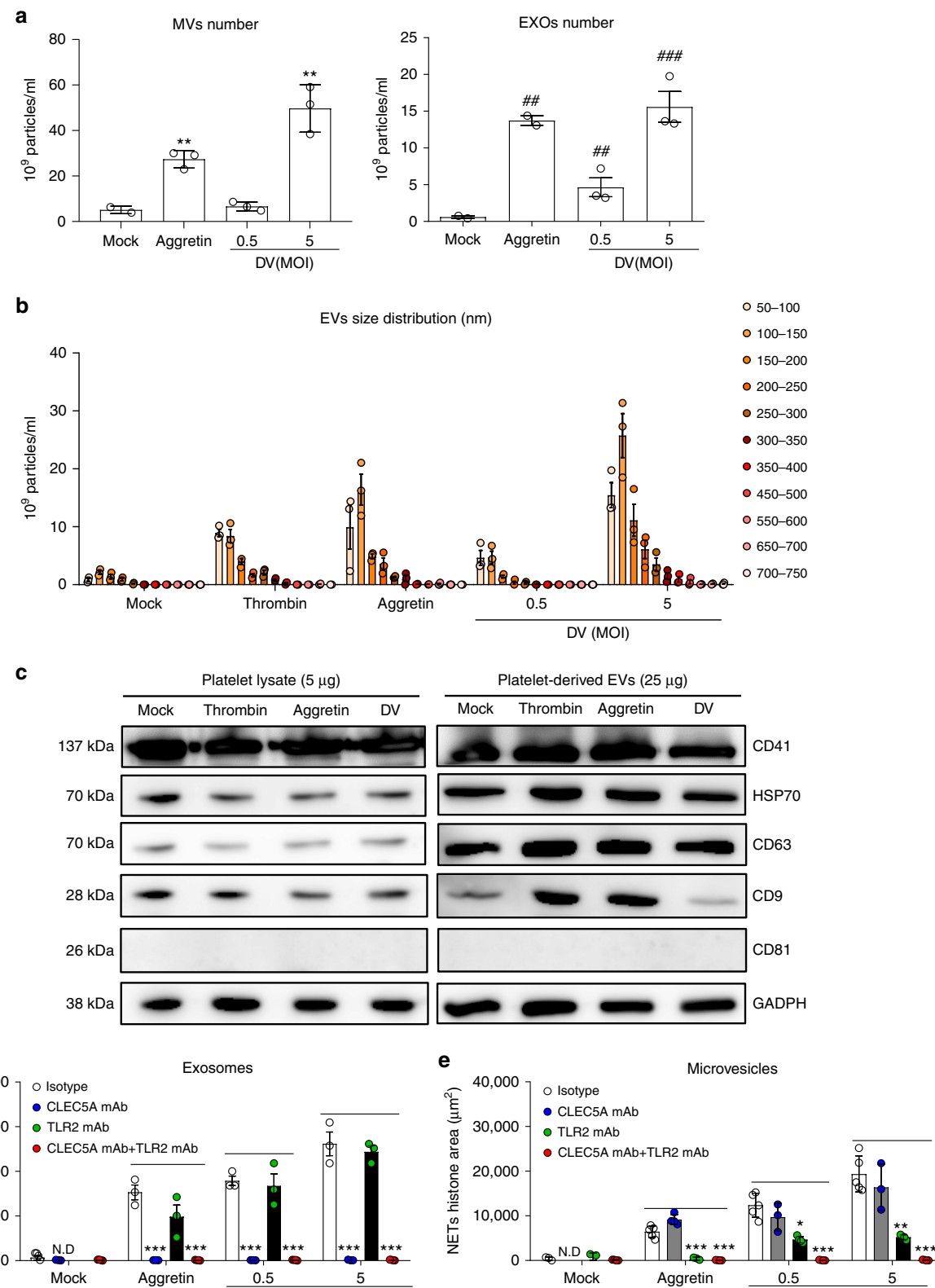

**Fig. 4** CLEC2-activated platelets release EVs to induce NETosis via TLR2 and CLEC5A. **a** and **b** Human platelets were incubated with DV (PL046), aggretin, or thrombin at 37 °C for 1 h, and microvesicles (MVs) and exosomes (EXOs) were harvested by sequential ultracentrifugation from supernatants, followed by nanoparticle tracking analysis (NTA). **c** PLT lysates and EVs from platelets were fractionated on SDS–PAGE and subjected to western blotting with anti-CD41, anti-HSP70, anti-CD63, anti-CD9, and anti-CD81 mAbs. **d** and **e** Neutrophils were pre-incubated with anti-CLEC5A mAb and anti-TLR2 mAb, followed by incubation with microvesicles ($1.5 \times 10^8$) and exosomes ($1.5 \times 10^8$) from aggretin-PLTs or DV-PLTs at 37 °C for 3 h. Data are mean ± SEM from at least three independent experiments. **$p < 0.01$, ***$p < 0.001$ (mock versus aggretin-MVs or DV-MVs); ##$p < 0.01$, ###$p < 0.001$ (mock versus aggretin-MVs or DV-EXOs) (Student's $t$-test). Source data are provided as Source Data file

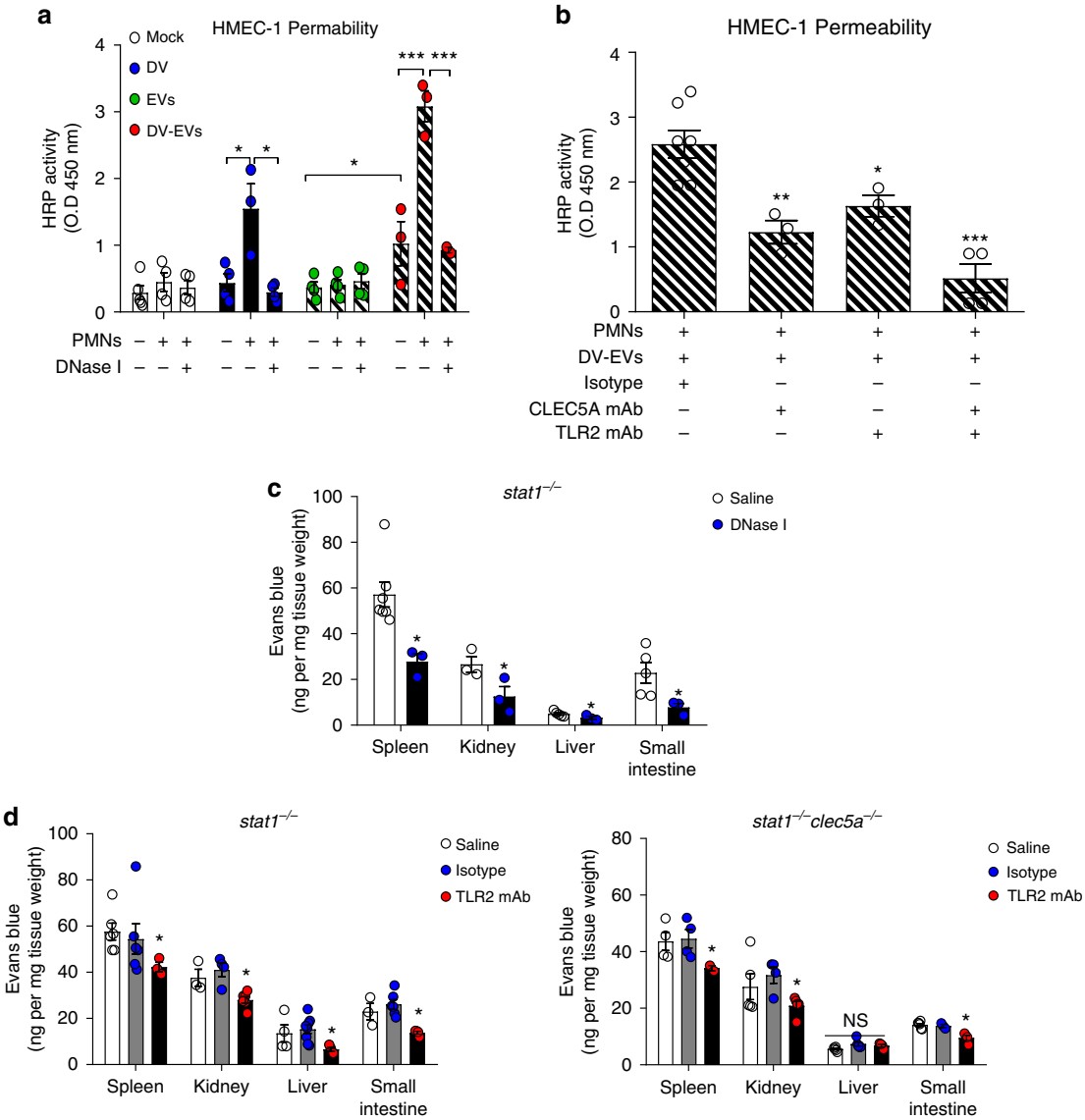

**Fig. 5** DV-induced NETs contribute to changes in vascular permeability. **a** and **b** Neutrophils were incubated with DV, EVs (from $8 \times 10^6$ platelets, including both MVs and EXOs), or DV-EVs (from $8 \times 10^6$ DV-activated platelets) and overlaid on HMEC-1 monolayers in presence or absence of DNase I (100 U/ml) at 37 °C for 3 h (**a**). Alternatively, neutrophils were pretreated with isotype control, anti-CLEC5A mAb, and anti-TLR2 mAb at 37 °C for 1 h, followed by incubation with DV or EVs (**b**). These assays used $1.5 \times 10^8$ EVs per experiment (**a** and **b**). Endothelial monolayer permeability was measured by determining HRP activity in media from the lower chamber. **c** $Stat1^{-/-}$ mice were challenged by intravenous administration of DV, in the presence or absence of DNase I, followed by injecting Evans blue (i.v.) at 1 h before sacrifice on day 5 post-infection ($n = 4$ in saline (vehicle) group, $n = 3$ in DNase I group). **d** $Stat1^{-/-}$ ($n = 6$ in saline and isotype control group, $n = 4$ in anti-TLR2 mAb group) and $stat1^{-/-}clec5a^{-/-}$ ($n = 5$ in saline group, $n = 4$ in isotype control and anti-TLR2 mAb group) mice were pre-treated with anti-TLR2 mAb or isotype control, followed by inoculation with DV (NGC-N) via intraperitoneal route. Evans blue was injected intravenously at day 5 post-infection, 1 h before mice were sacrificed. Plasma leakage of visceral organs was determined by measuring absorbancies at 610 nm for each organ. Data are mean ± SEM from at least three independent experiments. *$p < 0.05$, **$p < 0.01$, ***$p < 0.001$ (Student's $t$-test). Source data are provided as Source Data file

Fig. 7d). However, anti-TLR2 mAb increased the survival rate of $stat1^{-/-}clec5a^{-/-}$ mice to 90% after DV infection (right, Fig. 7d). These observations suggest that simultaneous blockade of TLR2 and CLEC5A can further reduce DV-induced inflammatory reactions and protect mice from DV-induced lethality.

## Discussion
Serum levels of EVs are regarded as valuable biomarkers to predict clinical outcomes of DV infection[21]. However, whether EVs contribute to the pathogenesis of dengue hemorrhagic shock had not been addressed prior to this study. Here, we have demonstrated that DV activates CLEC2 in platelets to release EVs, thereby enhancing NET formation and proinflammatory cytokine release via the activation of CLEC5A and TLR2 in neutrophils and macrophages. Simultaneous blockade of CLEC5A and TLR2 was found to increase the survival rates of mice infected with DV (90% survival), whereas we, and others[13], have shown that blockade of CLEC5A alone only provides partial protection against DV-induced lethality. Moreover, removal of NETs by DNase I in vivo had some protective effect in $stat1^{-/-}$ mice (35% survival) (left, Supplementary Fig. 3), supporting the

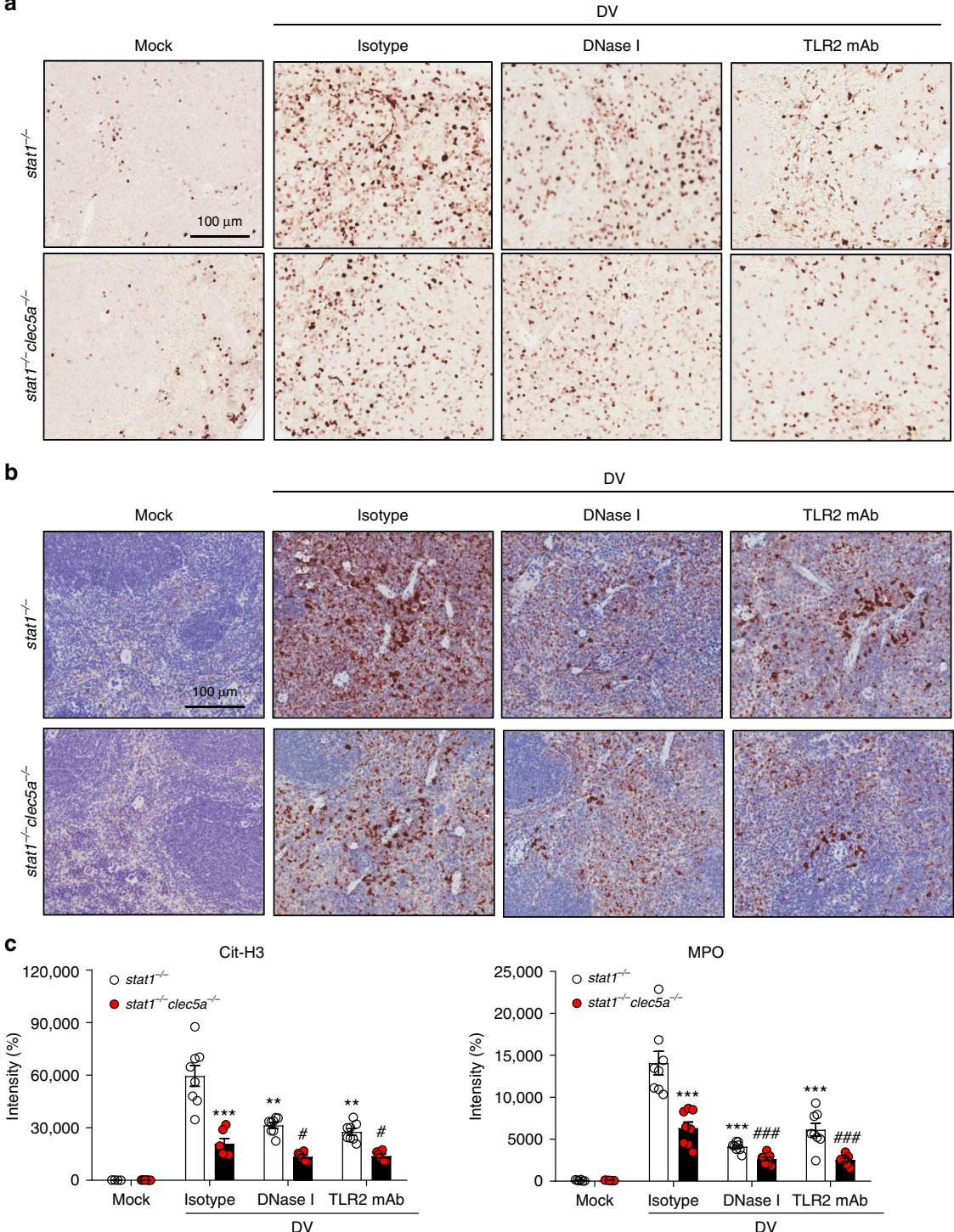

**Fig. 6** DV-induced NET formation in vivo occurs via CLEC5A and TLR2. **a** and **b** S*tat1*$^{-/-}$ and *stat1*$^{-/-}$*clec5a*$^{-/-}$ mice were pre-treated with isotype control mAb, DNase I, and anti-TLR2 mAb, followed by inoculation with DV (NGC-N) via intraperitoneal route. Spleens were harvested and fixed at day 5 post-infection for immunohistochemical staining using anti-citrullinated histone H3 (Cit-H3) mAb (**a**) or anti-MPO mAb (**b**) ($n = 3$ per group). **c** Cit-H3 and MPO were quantified using MetaMorph image analysis software. \*\**p* < 0.01, \*\*\**p* < 0.001 (compared with isotype control-injected *stat1*$^{-/-}$ mice); #*p* < 0.05, ###*p* < 0.001 (compared with isotype control-injected *stat1*$^{-/-}$*clec5a*$^{-/-}$ mice) (Student's *t*-test). Scale bar: 100 μm. Source data are provided as Source Data file

notion that NETs are involved in the complex interaction between platelets, macrophages, and neutrophils during DV infection. In contrast, DNase I did not enhance the survival of DV-challenged *stat1*$^{-/-}$*clec5a*$^{-/-}$ mice (right, Supplementary Fig. 3), which did not have obvious NET formation after DV

challenge. These observations suggest that DV-induced NET formation exerts a pathogenic effect by contributing to increased systemic vascular permeability during DV infection.

The role of virus-induced NETs in host defense against viral infections remains controversial[22,23]. NETs can protect the host

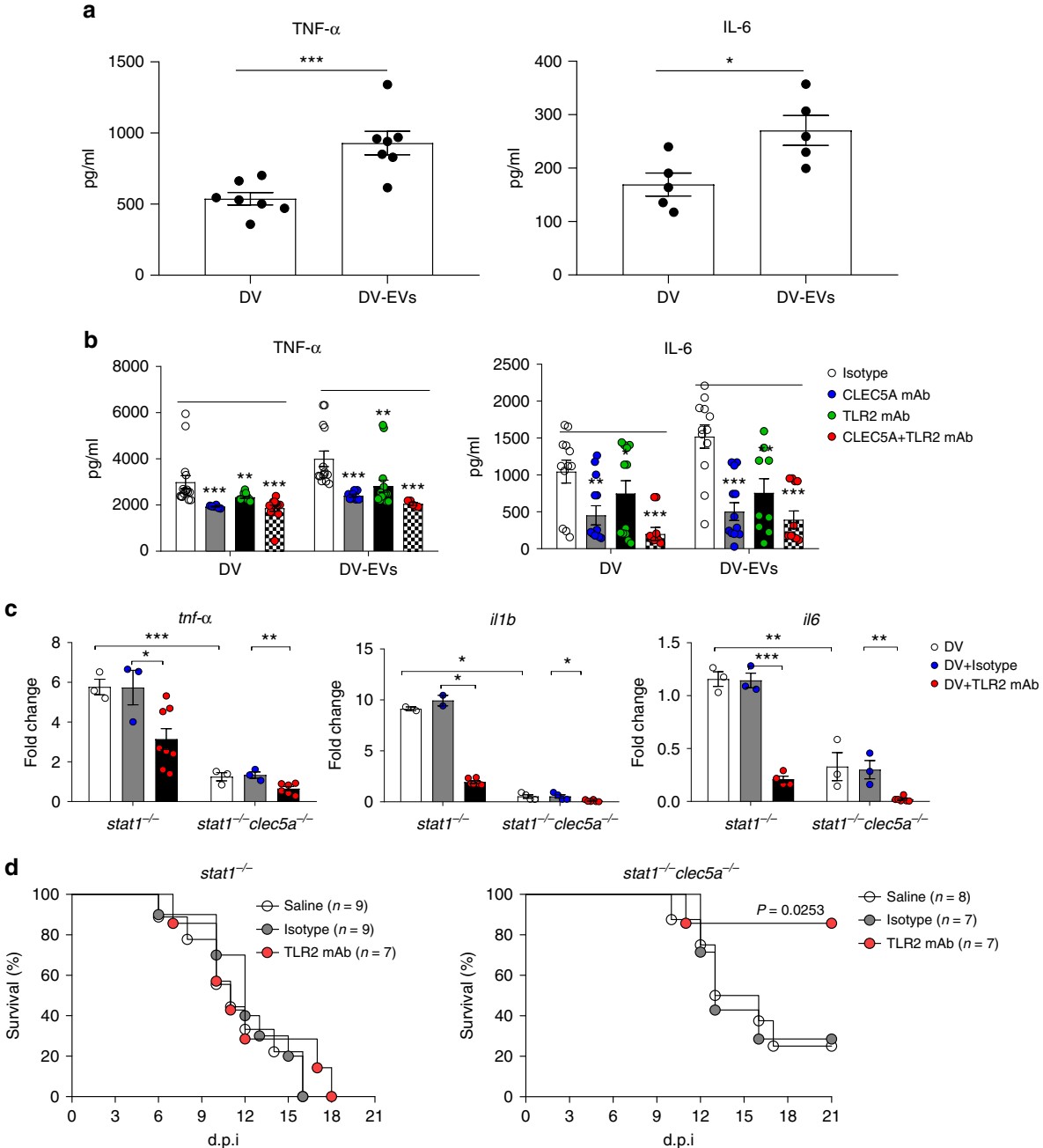

**Fig. 7** DV induces inflammatory response and lethality via CLEC5A and TLR2. **a** and **b** Human macrophages were incubated with DV or DV-EVs (including both MVs and EXOs) for 2 h at 37 °C, washed and then incubated for a further 24 h (**a**). Alternatively, cells were preincubated with anti-CLEC5A mAb and anti-TLR2 mAb for 1 h before addition of DVs (**b**). **c** Spleens of *stat1*$^{-/-}$ *and stat1*$^{-/-}$*–clec5a*$^{-/-}$ mice were collected at day 5 post DV-infection (as described in Fig. 6) and levels of TNF-α, IL-1β, and IL-6 mRNA were determined by RT-qPCR (n = 3 per group). **d** S*tat1*$^{-/-}$ mice (left panels, n = 9 in saline and isotype control group; n = 7 in anti-TLR2 mAb group) and *stat1*$^{-/-}$*clec5a*$^{-/-}$ mice (right panels, n = 8 in saline; n = 7 in isotype control group and anti-TLR2 mAb group) were challenged with a lethal dose of DV via intraperitoneal route, and isotype control mAb (100 μg) or anti-TLR2 mAb (100 μg) were injected intraperitoneally at day 0, 2, 4, and 6 post-infection. Survival rate was measured daily up to day 21 post-infection. Data are mean ± SEM from at least three independent experiments. *p < 0.05, **p < 0.01, ***p < 0.001 (Student's *t*-test). Source data are provided as Source Data file

from infection with HIV and pox virus via sequestration and neutralization of virus[24]. On the other hand, local NET deposition contributes to RSV-induced inflammatory reactions[25] and to IAV-induced lung injury[26]. Moreover, increased NET formation is associated with the severe form of DV infection—dengue hemorrhagic fever[27]. Therefore, the roles of NETs seem to depend on the complex interplay between individual viruses and host cells. Here, we have demonstrated that DV not only activates macrophages and neutrophils through stimulation of CLEC5A,

but also activates platelets to release EVs via CLEC2. Increased systemic vascular permeability contributes significantly to DV-induced mortality and enhanced NET formation is known to impair endothelial integrity and aggravate plasma leakage. This is consistent with our observation that, blockade of NET formation maintains vascular integrity and prevents increases in systemic permeability, thereby protecting mice from DV-induced lethality.

CLEC2 has been reported to play a critical role in hemostasis and thrombosis[16], and is required for efficient binding of HIV-1 to

platelets[28]. In this study, we demonstrate that CLEC2 is a novel PRR for DV, and that CLEC2-activated platelets release EVs, thereby enhancing NET formation and inflammatory reactions, following DV infection. While EVs from tumor cells transfer nucleic acids and proteins to target cells to suppress the immune system and facilitate tumor growth[18], we have shown that DV-EVs activate neutrophils and macrophages to enhance NET formation and proinflammatory cytokine release. In addition to DV, we have also found that IAV (H5N1) activate platelets to release EVs via CLEC2, thereby enhancing NET formation (Supplementary Fig. 4). This observation further supports a critical role for CLEC2 and platelets in the pathogenesis of viral infection. Interestingly, LPS-activated and thrombin-activated platelets have similar functions to DV-activated and aggretin-activated platelets, i.e. enhancing NET formation via CLEC5A and TLR2 (Supplementary Fig. 5). Thus, both CLEC2-dependent and CLEC2-independent pathways contribute to enhanced NET formation in response to EVs from activated platelets. Recently, we reported that CLEC5A and TLR2 were co-activated by *Listeria* and *Staphylococcus* to induce NET formation and inflammatory cytokine release[14]. In this study, we further demonstrated that activated platelets can release EVs to enhance NET formation via CLEC5A (DV-EXOs) and TLR2 (DV-MVs). Thus, co-activation of CLEC5A and TLR2 seems to be a common pathway leading to NET formation in response to pathogens and other danger signals.

It has been demonstrated that CLEC5A colocalizes with mannose receptor (MR) and DC-SIGN in the presence of DV, where both MR and DC-SIGN display higher binding affinities for DV compared to CLEC5A, and DV-induced CLEC5A activation is dependent on DC-SIGN and MR[29]. Based on imaging analysis, the authors concluded that CLEC5A, DC-SIGN, and MR form a hetero-multivalent receptor complex upon engagement with DV[29] that has high affinity for the virus and enables signaling via CLEC5A. While we have demonstrated that DV-mediated activation of platelets occurs via CLEC2, Watson et al. reported that interaction of CLEC2 with DV was not detectable[30]. However, DV has been shown to bind platelets via DC-SIGN[31], suggesting that DV could activate CLEC2 via binding to a DC-SIGN/CLEC2 hetero-multivalent receptor complex, despite having low affinity for CLEC2.

We have found that DV-EVs purified by ultracentrifugation have low levels of contamination with DV as revealed by plaque assay and PCR (~150–200 pfu/ml, Supplementary Fig. 6). However, low titer DV did not induce NET formation, and a DV neutralizing antibody did not abolish DV-EVs-induced NET formation. All these observations suggest that NET formation is mainly induced by EVs from activated platelets, and not by the contaminating DV in DV-EVs preparations.

We performed mass spectrometry to identify components of DV-EXOs and DV-MVs capable of enhancing NET formation. Among the 234 peptides detected by mass spectrometry, tubulin beta 1 chain (TUBB1), guanine nucleotide-binding protein (GNG3), tribbles homolog 1 (TRIB1), vinculin (VCL), coagulation factor XIIIa chain (F13A1), and calnexin (CANX) were upregulated by both aggretin and DV (Supplementary Table 1). It has been shown that cytoskeleton components[32] and chaperones[33] are endogenous danger signals capable of activating innate immunity receptors. Cytoskeletal F-actin has been identified as a ligand for C-type lectin 9A (CLEC9A)[34], while the chaperone HSP70 is a ligand for TLR2 and TLR4[35]. Thus, it would be interesting to determine whether TUBB1, VCL (components of cytoskeleton), and CANX (an endoplasmic reticulum chaperone) are able to activate CLEC5A and TLR2. In addition to protein component, it is becoming apparent that glycosylation is an important factor in EVs function[36]. In this regard, it would be interesting to investigate whether glycans on DV-MVs and DV-EXOs contribute to the activation of CLEC5A and TLR2.

A high-density core region was found in DV-activated MVs, but not in aggretin-activated MVs (Supplementary Fig. 7). Mass spectrometry analysis, identifed 12 peptides upregulated by DV, but not by aggretin; these corresponded to proteins including the homeobox protein PKNOX2, isoform 20 of cAMP-responsive element modulator (CREM), POTE ankyrin domain family member I, cell division control protein 42 (CDC42) homolog, protein SSUH2 homolog, PRAME family member 9, ARF GTPase-activating protein (GIT2), LPS-responsive and beige-like anchor protein (LRBA), protein crumbs homolog 1/3 (CRB1/3), threonine aspartase 1 (TASP1), probable ubiquitin carboxyl-terminal hydrolase 4 (MINDY-4), and isoform 2/3 of calmodulin-regulated spectrin-associated protein 2 (CAMSAP2). It will be interesting to examine whether these proteins are present in the core region of DV-MVs (Supplementary Fig. 7) and whether they have any immunomodulatory functions.

While previous reports showed that platelets became apoptotic after incubation with DV for 6–96 h[37,38], we did not observe platelet apoptosis after incubation with DV PL046 (human) or NGC-N strains for short period (1–3 h). Instead, we found that DV activated CLEC2 in platelets to release EVs, which further enhanced DV-induced NET formation and inflammatory reactions. This observation is in accord with a previous report that EV levels correlate with disease severity in dengue patients[21], and supports the argument that neutrophil–platelet interactions contribute to enhanced inflammatory reactions during DV infection[39].

This study further illustrates the critical role of C-type lectin receptors and TLR2 in platelet–leukocyte interactions during viral infections (Fig. 8), and supports the notion that platelets are central to host immune responses to virus[40]. DC-SIGN and CLEC2 in platelets capture human immunodeficiency virus to promote virus spreading[28], while DC-SIGN captures DV to facilitate virus replication in platelets[31]. In addition, DV can activate CLEC2 to trigger the NALP3 inflammasome and induce IL-1β release from platelets[9,38], thereby enhancing proinflammatory cytokine release and NET formation via CLEC5A and TLR2. Histone 2A has been shown to activate platelets and enhance inflammatory reactions in patients infected with DV[41]; therefore, DV-induced NETs could further amplify neutrophil–platelet interaction via the release of histone 2A and other nuclear components to activate platelets. Moreover, IAV H5N1 also stimulates CLEC2 to produce EVs and enhance NET formation. These observations suggest that activation of CLEC2 and CLEC5A/TLR2 in platelets and leukocytes by DV and IAV contributes significantly to disease severity, and simultaneous blockade of CLEC5A and TLR2 has a great potential to treat DV, IAV, and other viral infections in the future.

## Methods

**Reagents and antibodies.** Culture media/supplements were purchased from Invitrogen GIBCO. Chemical reagents were from Sigma. Granulocyte–macrophage-colony-stimulating factor (GM-CSF) was from R&D system. Antibodies for flow cytometry were purchased from BioLegend: anti-human CD41 antibody (#303710), anti-mouse CD41 antibody (#133914), anti-human CD63 antibody (#353006), anti-mouse CLEC2 antibody (#146103), anti-human/mouse TLR2 antibody (#153002); anti-human CD62p antibody (#304906); anti-mouse CD62p antibody (#561923). Mouse anti-human CLEC2 antibody (clone AYP1) for immunofluorescence staining was provided by Steve P Watson (University of Birmingham, UK); other primary antibodies for immunostaining were mouse anti-human/mouse histone H1 antibody (#sc-8030; Santa Cruz), rabbit anti-human/mouse myeloperoxidase (MPO) antibody (#AF3667; R&D Systems). Secondary antibodies (The Jackson Laboratory) were: goat anti-rabbit (H+L) TRITC-conjugated antibody (#111-025-045), donkey anti-goat Alexa488-conjugated antibody (#705-545-147), goat anti-mouse (H+L) Alexa488-conjugated antibody (#115-545-146), rabbit anti-goat (H+L) HRP-conjugated antibody (#305-035-045), donkey anti-rabbit HRP-conjugated antibody (#711-035-152). Antibodies for Western blotting were: anit-CD9 (#EXOAB-CD9A-1), anti-CD63 (#EXOAB-CD63A-1), anti-CD81 (#EXOAB-CD81A-1), and anti-HSP70 antibody

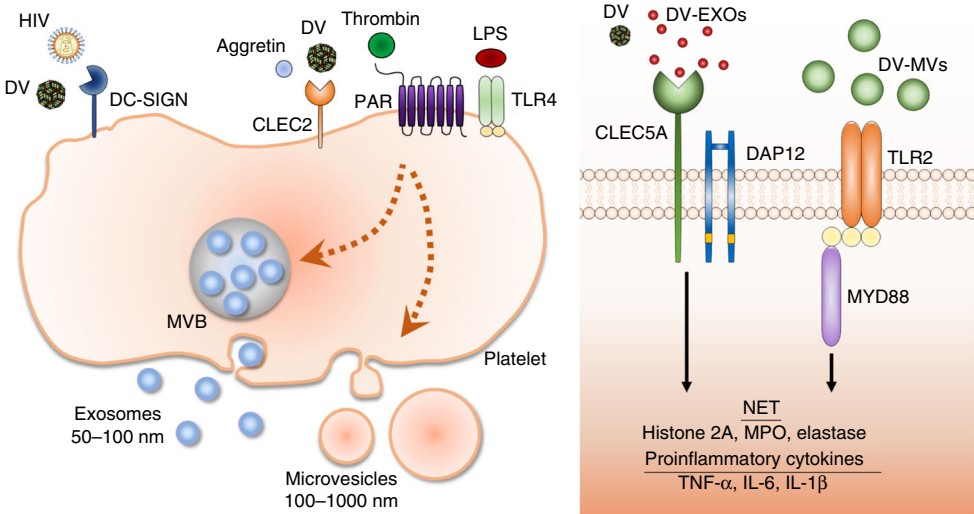

**Fig. 8** EVs play critical roles in the pathogenesis of viral infectons. DV activates macrophages and neutrophils via CLEC5A. DV also stimulate CLEC2 on platelets to promote release of EVs (DV-EXOs and DV-MVs), which further enhance the secretion of proinflammatory cytokines and NET formation in neutrophils via CLEC5A and TLR2

(#EXOAB-Hsp70A-1) all from System Biosciences; anti-CD41 (#13807S; Cell Signaling), anti-GAPDH (#MAB374; Millipore). Reagents for IHC staining were: anti-citrullinated Histone H3 (#NB100-57135; Novus), anti-human/mouse MPO antibody (#AF3667; R&D Systems). ELISA kits to detect TNF-α (#DY410) and IL-6 (#DY406) were purchased by R&D Systems.

**Isolation of human primary cells.** Human platelets and neutrophils were isolated from peripheral blood of drug-free healthy donors. Platelets were harvested by centrifugation at $230 \times g$ for 15 min at 24 °C, followed by centrifugation at $1000 \times g$ for 10 min at 24 °C. Pellets were re-suspended with Tyrode's buffer (Sigma, T2397) containing 1/6 volumes of ACD anticoagulant and 1 μM PGI2, further centrifuged at $1000 \times g$ for 10 min at 24 °C and resuspended in Tyrode's buffer containing 1 μM PGI2 and 0.04 U/ml apyrase, then kept on an orbital shaker (50 rpm) at 24 °C. Before use, samples were subjected to centrifugation at $1000 \times g$ for 10 min at 24 °C and resuspended in Tyrode's buffer ($3.2 \times 10^8$ platelets/ml). Neutrophils were isolated using Ficoll-Paque (GE Healthcare, 45-001-748) gradient centrifugation at $500 \times g$ for 15 min, and pellets were resuspended in red blood cell lysis buffer, then washed once with saline ($300 \times g$, 5 min) before being resuspended in RPMI containing 10% (v/v) autologous serum. To obtain $CD14^+$ monocyte-derived macrophages, $CD14^+$ monocytes were isolated from human white blood cell concentrates (Taipei Blood Center,Taiwan) using anti-CD14 microbeads (Miltenyi Biotec, 130-050-201)[8]. $CD14^+$ cells were cultured in RPMI containing 10% (v/v) FBS and 10 ng/ml human GM-CSF for 7 days at 37 °C and 5% $CO_2$. The protocol was approved by the Human Subject Research Ethics, Academia Sinica (Taiwan, number AS-IRB-BM-16074).

**Mice.** C57BL/6J mice were purchased from the National Laboratory Animal Center (NLAC, Taiwan). $clec5a^{-/-}$, $stat1^{-/-}$, and $stat1^{-/-}clec5a^{-/-}$ mice were generated in house[8,12,13], while $tlr2^{-/-}$ and $clec5a^{-/-}tlr2^{-/-}$ mice were generated as described in our previous work[14]. Clec2fl/fl mice were a generous gift from Dr. Steve P. Watson (University of Birmingham, UK), and PF4-Cre mice were the generous gift from Dr. Ching-Ping Tseng (Chang-Gung University, Taiwan). Clec2fl/fl mice were bred with PF4-Cre mice to generate platelet-specific $clec2^{-/-}$ mice[42]. All mice were bred and maintained in pathogen-free conditions at Academia Sinica SPF animal facility (AS core) and The Laboratory Animal Center of National Defense Medical Center. Animal experiments were approved by the Institutional Animal Care and Use Committee (IACUC) at AS core (protocol ID 16-12-1033). In this study, we used 8–12-week-old male and female mice, and we did not observe any significant difference in response to virus infection between males and females.

**Isolation of murine primary cells.** To isolate murine platelets, blood samples were collected from the facial vein. The platelet-rich plasma was centrifuged at $1000 \times g$, and platelets were resuspended in Tyrode's buffer ($3.2 \times 10^8$ platelets/ml) before use. To isolate murine neutrophils, mouse bone marrow cells were collected from femurs and tibias. After washing and RBC lysis, samples were resuspended in 45% percoll solution and laid on percoll gradient solution (with 81%, 62%, 52% percoll from bottom to top), followed by centrifugation at $1000 \times g$ for 30 min. Bone marrow-derived neutrophils were collected from layer 3 of the gradient solution.

Cells were washed with HBSS solution and resuspended in RPMI containing 10% (v/v) FBS.

**Cell lines.** Mosquito C6/36 cells (ATCC, CRL-1660) were cultured in RPMI medium supplemented with 10% FBS at 28 °C and 5% $CO_2$. Hamster BHK-21 cells (ATCC, CCL-10) were cultured in RPMI medium supplemented with 5% FBS, while human HMEC-1 cells (ATCC, CRL-3243) were cultured in M200 medium (Gibco, #M200500) supplemented with low serum growth supplement (LSGS) (Gibco, #S00310). These two cell lines were cultured at 37 °C and 5% $CO_2$. Trypsin–EDTA was used to detach cells from culture plates.

**Virus strains and propagation.** DV serotype 2 strains PL046 and New Guinea C-N (NGC-N) were propagated in C6/36 cells. Virus titers were determined by plaque assay in BHK-21 cells. Human and mouse blood cells were incubated with PL046 strain and NGC-N strain, respectively.

**In vitro stimulation of platelets and collection of platelet-derived EVs.** Human platelets were incubated with DV (MOI = 0.5 or 5) or aggretin (3, 10, or 100 nM) for 1 h at 37 °C, and the expression of CD62p and CD63 was determined by flow cytometry. Samples were subjected to centrifugation at $1500 \times g$ for 10 min at 24 °C to harvest the EVs-enriched supernatants.

**Analysis of platelet-derived EVs by flow cytometry.** Human platelets and $CD41^+$ EVs were fixed with ThromboFix reagent (Beckman Coulter, #6607130) for 1 h at room temperature according to the vendor's instruction, followed by incubation with APC-conjugated anti-CD41 mAb, FITC-conjugated anti-CD63 mAb, and PE-conjugated anti-CD62p mAb for 30 min at room temperature before being subjected to flow cytometry analysis. The numbers of EVs derived from activated platelets were extrapolated from EV counts obtained for 10,000 $CD41^+$ platelets. The gating strategy is shown in Supplementary Fig. 8.

**NTA of platelet-derived MVs and EXOs.** DV (PL046)-containing supernatant was centrifuged at $100,000 \times g$ for 1 h at 4 °C to remove endogenous EVs in the culture medium before incubation with freshly purified human platelets (MOI = 0.5) at 37 °C for 1 h. Samples were then centrifuged at $1500 \times g$ for 15 min at room temperature to harvest the EVs-enriched supernatant, which was further subjected to centrifugation at $20,000 \times g$ for 40 min at 4 °C, and the pellets were resuspended in 1 ml of Tyrode's buffer to harvest microvesicles. The supernatant was further subjected to centrifugation at $100,000 \times g$ for 1 h at 4 °C, and the pellets were resuspended in 1 ml of Tyrode's buffer to harvest exosomes. The particle size and concentration of microvesicles (100–1000 nm) and exosomes (<100 nm) were determined by NTA (NanoSight NS300).

**Stimulation and inhibition of platelets, neutrophils, and macrophages.** Human neutrophils ($8 \times 10^5$) were seeded onto poly-L-lysine-coated glass coverslips and incubated at 37 °C for 1 h, followed by incubation with DV (MOI = 5) in the presence or absence of autologous platelets ($8 \times 10^6$) at 37 °C for 3 h. Alternatively, neutrophils were incubated with EVs (from $8 \times 10^7$ platelets) at 37 °C for 3 h. For blocking assays, neutrophils were pre-incubated with anti-human CLEC5A mAb

(100 μg/ml), anti-TLR2 mAb (100 μg/ml), or isotype control for 1 h at 24 °C, while platelets were pre-treated with anti-human CLEC2 mAb (2.5 μg/ml, clone AYP1) for 15 min at 24 °C. Human macrophages ($1 \times 10^5$ cells) were incubated with DV (MOI = 10) or DV-EVs (from $1 \times 10^7$ DV-PLTs) for 2 h at 37 °C, followed by washing with RPMI-1640 medium to remove DV and DV-EVs before incubation with fresh RPMI containg 10% (v/v) FBS for 24 h. For blocking assay, macrophages ($1 \times 10^5$) were preincubated with anti-CLEC5A mAb (10 μg/ml, clone 3E12A2), anti-TLR2 mAb (10 μg/ml), or isotype control at 37 °C for 1 h before incubation with DV or DV-EVs (from $1 \times 10^7$ DV-PLTs).

**Cytokine measurement**. The concentrations of cytokines in media from human macrophages were measured by ELISA according to the manufacturer's instructions.

**Determination of citrullinated histone and MPO by immunohistochemistry**. To detect citrullinated histone and MPO in the spleen, specimens were fixed in 10% formalin for 24 h. Paraffin-embedded spleen sections were incubated at 70 °C for 1 h and deparaffinized in Xylene for 10 min twice. Samples were incubated in boiling citric acid (0.01 M, pH. 6.0) for 30 min, then cooled to room temperature for antigen retrieval. Samples were incubated with anti-citrullinated histone antibody (1:100) or anti-MPO antibody (1:100) at 4 °C overnight, then incubated with HRP-conjugated secondary antibody (1:100) for 1 h at room temperature; HRP activity was determined using the Pierce® Peroxidase Detection Kit (Thermo Fisher Scientific, #36000). Samples were counterstained with hematoxylin, and DAB signal was detected and quantified using Aperio ImageScope V9 and MetaMorph softwares.

**Determination of NET structure by immunofluorescence staining**. NET structure was determined by detection of citrullinated histone and MPO using confocal microscope[14]. In brief, neutrophils were suspended in RPMI with 10% (v/v) autologous serum and seeded on coverslips for 2 h. Cells were infected with either DV or EVs and then fixed with 4% paraformaldehyde. Fixed cells were subjected to membrane permeabilization using 0.5% Triton X-100 in PBS, then blocked with 3% bovine serum albumin in PBS for 1 h at room temperature, and incubated with primary antibodies against anti-histone and MPO (1:100) at 4 °C overnight. Secondary antibodies conjugated with Alex488 or TRITC (1:100) were then added for 1 h at room temperature, and Hoechst 33342 was used as a counter stain. NET images were captured under a confocal microscope and analyzed with Leica Application Suite X software. To observe NETs in situ, mouse spleens were dissected, mounted in OCT and then frozen at −80 °C; 10 μm sections were used for immunofluorescence staining. Slides were fixed with cold acetone at 4 °C for 10 min, air dried for 30 min and then incubated with Hoechst 33342 and primary antibodies to histone and MPO (1:100) followed by fluorochrome-conjugated secondary antibodies (1:100). Images were collected with a Leica confocal microscope with white light laser system (TCS-SP5-MP-SMD) and analyzed using Leica Application Suite X software.

**RNA isolation and real-time PCR**. Mouse spleens were homogenized in 700 μl TRIzol reagent and RNA was isolated with TriRNA pure kit according to the vendor's instructions. For reverse-transcription, the first-strand cDNA was synthesized using the RevertAid First Strand cDNA Synthesis Kit. Primer sequences for mouse TNF-α: forward 5′-gcctcttctcattcctgcttg-3′, reverse 5′-ctgatgagagggaggccatt-3′; mouse IL-1β: forward 5′-ggagaaccaagcaacgacaaaata-3′, reverse 5′-tgggggaactctgcagactcaaac-3′; mouse IL-6: forward 5′-gaggataccactcccaacagac-3′, reverse 5′-aagtgcatcgttgttcataca-3′; mouse GAPDH: forward 5′-ggaggaacctgccaagtatg-3′, reverse 5′-tgggagttgctgttgaag-3′. Real-time PCR conditions comprised incubation at 95 °C for 5 min, followed by 30 cycles of 15 s at 95 °C, 30 s at 58 °C, and 30 s at 72 °C. The expression level of each gene was normalized to GADPH expression.

**Endothelial cell permeability assay**. Permeability of human dermal microvascular endothelial cell (HMEC-1) monolayers was determined by measuring the trans-endothelial passage of HRP after incubation with supernatants of DV-infected macrophages. Briefly, HMEC-1 ($2 \times 10^5$) were seeded in 24-well collagen-coated and fibronectin-coated transwells (6.5 mm diameter, 0.4 μm pore size) and cultured in Medium 200 with LSGS. On day 1 post-plating, neutrophils ($8 \times 10^5$) were laid on the monolayer of HMEC-1 and incubated with DV (MOI = 5) or DV-EVs (from $8 \times 10^6$ DV-PLTs) for 3 h. HRP (0.5 μg in 10 μl) was added to the upper chamber of the transwell for 15 min before harvesting medium from the lower chamber (10 μl). HRP activity was determined by incubating the media with TMB (3,3′,5,5′-tetramethylbenzidine) for 20 min, followed by determination of absorbancy at 450 nm after the addition of sulfuric acid to stop the reaction.

**Quantitation of vascular permeability change**. Mice were inoculated intraperitoneally with DV2 (New Guinea C-N) at $2 \times 10^5$ PFU/per mouse in 100 μl of saline. DNase I (4KU) and anti-mTLR2 mAb (BioLegend, #153002) or isotype control (100 μg) were injected intraperitoneally at days 0, 2, 4, 6 after DV infection. Evans blue (150 μl of 0.5% Evans blue) was administered intravenously into DV-challenged mice at day 5. At 1 h after injection, mice were sacrificed and organs

were harvested. Organs were incubated in formamide (2 ml) at 55 °C for 2 h, and Evans blue was quantified by measuring absorbance at 610 nm.

**Quantification of NET formation**. NET formation was quantified by measuring histone area[14]. Neutrophils were seeded onto coverslips and processed as described above for determination of NET structure. Images were collected using a Leica confocal microscope and processed using Leica Application Suite X and MetaMorph software. Briefly, NETs were visualized in at least five random fields (×40 magnification), signal intensity of histone per field was individually measured and the pixels count for each image was converted into area (μm²) using a calibration unit (0.8333). Mean histone area (μm²) was determined from the five independent fields.

**Western blot**. Platelet lysates (5 μg) and platelet-derived EVs (25 μg) were fractionated on SDS–PAGE under non-reducing conditions before transfer onto PVDF membrane (Pall Corporation). The blots were probed with mAbs to detect CD markers and HSP70 (1:1000), followed by incubation with peroxidase-conjugated goat anti-rabbit antibody (1:20,000) according to the manufacturer's instruction (System Biosciences) and development with the SuperSignal® West Femto Maximum Sensitivity Substrate kit (Thermo Scientific). All uncropped and unprocessed scans are supplied in the source data labeled as Fig. 4c.

**Mass spectrometry analysis of platelet-derived EVs**. Proteomic analysis was performed by the in house Mass Spectrometry Core Facility at the Genomic Research Center, Academia Sinica. EV samples were purified by ultracentrifugation as described in "Nanoparticle tracking analysis' above. Platelet-derived MVs and EXOs were digested with trypsin before being subjected to tandem mass spectrometry analysis using the LTQ Orbitrap XL mass spectrometer (Thermo Fisher Scientific Inc.).

**Transmission electron microscopy**. Samples were fixed with 1% (w/v) glutaraldehyde in PBS for 10 min, then loaded on a carbon/formvar-coated grid (5 μl each sample) and dried under vacuum for 20 min. Samples were incubated with 2% uranyl acetate (10 μl) for 30 s, then dried under vacuum for 20 min. The ultrastructure of each sample was observed by a transmission electron microscopy (FEG-TEM, FEI Tecnai G2 TF20 Super TWIN).

**Statistics**. All data were analyzed using GraphPad Prism software (Version 5.0) and are presented as mean ± SEM. For two group comparisons, significance was determined using a two-tailed non-paired $t$-test. For multiple group comparisons, one-way ANOVA with the Bonferroni post hoc test was used for parametric data; Kruskal–Wallis analysis with the Dunn's post hoc test was used for nonparametric data. Survival rate was assessed using Kaplan–Meier analysis. $p$-Values < 0.05 were considered significant.

**Reporting summary**. Further information on research design is available in the Nature Research Reporting Summary linked to this article.

## Data availability

The source data underlying Figs. 1, 2, 3, 4a, 4b, 4d, 4e, 5, 6c, 6d, 6d, 7b and 7c are provided as a Source Data file. Data that support this study are available within the main text and supplementary files, or from the corresponding author upon reasonable request.

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

## Acknowledgements

This work was supported by Academia Sinica (106-2101-01-11-01、 107-2101-01-18-03, AS-TP-106-L11, AS-TP-106-L11-1), and Ministry of Science and Technology (MOST 107-2321-B-001-015, MOST 106-2320-B-001-023-MY3), and Summit and Thematic Research Projects (Academia Sinica), and VGH, TSGH, AS Joint Research Program (VTA107-V2-1-2). We thank the Taiwan Mouse Clinic (TMC) and Transgenic Mouse Model Core Facility (TMMC), which are funded by the National Research Program for Genomic Medicine (NRPGM) at the National Science Council of Taiwan, for technical support in histopathology experiments. We thank the Mass Spectrometry Core Facility of Genomic Research Center (Academia Sinica) for technical support in mass spectrometry analysis. We thank the Imaging Core Facility in the Institute of Cellular and Organismic Biology (Academia Sinica) for TEM image processing. We thank Dr. Thierry Burnouf, Dr. Natalie Chou, Mrs. Li-Wen Lo, Ms. Tzu-Yun Hsu, Ms. Yi-An Chen, Mr. Heng-Yu Su, and Dr. Chun-Mei Hu for technical assistance. We are grateful to Dr. Ching-Ping Tseng (Chang-Gung University) for kindly providing the PF4-Cre mice to support this work. Special thanks go to Dr. Steve P. Watson who provided the antagonistic anti-human CLEC2 mAb (clone AYP1) and floxP-CLEC2-floxP mice. We also thank Bio-Legend for providing the antagonistic anti-TLR2 mAb (TLR2 mAb, clone QA16A01).

## Author contributions

P.-S.S. designed and performed the experiments, analyzed the data and wrote the paper; T.-F.H. provided reagents and technical support; S.-L.H. was involved in experimental design, data analysis and manuscript writing.
