## [Peer Review File · Nature Communications]

Reviewers' comments:

Reviewer #1 (Remarks to the Author):

The manuscript entitled "CLEC2-CLEC5A/TLR2 axis is critical in dengue virus-induced inflammation and lethality" by Sung et al. proposed the novel axis of CLEC2-CLEC5A/TLR2. The authors proposed a novel CLEC2-CLEC5A/TLR2 axis as a critical signaling pathway for dengue virus-induced inflammation. The authors also mentioned that the simultaneous blockade of CLEC5A and TLR2 may reduce dengue virus-induced inflammation and lethality. Although the study is potentially important from the clinical point of view, this reviewer feels that the manuscript would be much benefitted if the authors provide additional clarifications to following concerns.

Comments

- 1) There is no abstract.
- 2) No direct evidence is provided for the description "CLEC2-CLEC5A/TLR2 axis" throughout the manuscript.
- 3) It would be better to improve the quality of Figure 4A.
- 4) Figure 4D is missing in the current manuscript.
- 5) The authors may want to use Stat1-/-TLR2-/- mice to clarify the requirement of TLR2 in the infection experiment.
- 6) The each value in Figure 7B should be presented as actual concentration rather than fold change.

Reviewer #2 (Remarks to the Author):

In this paper the authors are interested in understanding how Dengue virus (DV) activates platelets and leukocytes and whether DV-induced leukocyte-platelet interaction contributes to disease severity. In this paper the authors show that DV activates a molecule called CLEC5A in neutrophils to induce NETosis which is further enhanced by the presence of platelets. It was already shown that CLEC5A can interact directly with the dengue virion and induce pro-inflammatory cytokine expression (Chen et al., Nature 2008). Moreover anti-CLEC5A monoclonal antibodies inhibit DV-induced plasma leakage, and vital-organ haemorrhaging reducing the mortality of DV infection by about 50% in STAT1-deficient mice. The authors suggest that DV activates CLEC2 in platelets to release extracellular vesicles (EVs), which further enhance NET formation and pro-inflammatory cytokine production via activating CLEC5A and TLR2 in both macrophages and neutrophils. Blocking CLEC2-CLEC5A/TLR2 axis may be a novel strategy to reduce tissue damage of patients suffering from severe DV infection.

This is a beautiful story with clearly presented figures and the content is potentially highly relevant for our understanding of DV pathogenesis. While the data is compelling and demonstrates a roles for a CLEC-2 platelet and CLEC5-neutrophil axis in NET formation, I have several questions about the mechanistic evidence that may need addressing.

Major concerns

- The authors claim that DV-activated platelets require CLEC2 to generate EVs that in turn induce NETosis via CLEC5. I am not convinced yet that alternative activation of platelets will not lead to similar NET formation. Could the authors show that CLEC2-independent activation of platelets (PMA/von Willebrand factor/PAR-1 agonists?) have simialr effects?

- The authors show that DV itself also activates NET formation (Fig 2A) which is increased by adding CLEC2-activated PLTs. It is unclear though how the authors separate DVs from the DV-activated PLTs and MVs. One may assume that the MVs and EXOs preparations will be contaminated with intact DVs. Can the authors rule out this possibility? How are the amount of material, PLTs, MVs and EXOs compared?

- While CLEC2 mAb and CLEC2^{-/-} platelets lose their ability to cause NET formation, it remains unclear what the critical molecule(s) on the platelets and or EVs is/are that support (CLEC5-dependent) NET formation. Can the authors identify critical molecules on the PLTs/MVs or EXOs that activate the NET formation? I'm asking because one may hypothesize that CLEC2 activation of PLTs may induce selected molecules for secretion via EVs.

Minor

What is the concentration of isotype control in Fig 2A?'

The western blots of the MVs and EXs are not extremely convincing, in particular CD63 bands are suboptimal what is the predicted size maybe use non-reducing conditions. In addition, other EV protein markers such as HSP70 or CD81 and TSG101 or ALIX should be shown.

Finally it would behoove the authors to show Electron Microscopy images of the PLT, MV and EXO preparations to get an idea of their composition/purity.

Reviewer #3 (Remarks to the Author):

Sung and colleagues analyzed the involvement of CLEC-2 and CLEC-5 in DENV-induced platelet activation and platelet-mediated NET extrusion. Similar to platelets, platelet-derived extracellular vesicles were also able to increase NET extrusion. The participation of platelet CLEC-2 and neutrophil CLEC-5 and TLR-2 in NET extrusion in vitro is very convincing. The participation of CLEC-5 and TLR-2 in NET extrusion and increased vascular permeability is also very convincing in vivo. However, there is no evidence for the participation of platelets or platelet-CLEC-2 in the release of NET, increased vascular permeability or mortality in vivo. Even though the data presented here are of interest, some concerns must be addressed.

Major issues:

1- The authors mention platelet-specific CLEC-2 KO mice (CLEC2^{flox}/PF4^{cre}) in material and methods (line 337). Were platelet-specific CLEC-2 KO mice protected from DV infection? Were they susceptible to platelet activation, NET extrusion in vivo and increased vascular permeability? In addition, platelet depletion experiments would be useful to clarify the importance of platelets to NET extrusion and vascular damage in vivo.

2- Identification of extracellular vesicles (EVs) as microparticles (MPs) or exosomes is a crucial aspect of the author's conclusions. Importantly, the authors do not present any data on characterization of platelet-derived MPs and exosomes (size, number, surface markers, ultrastructure...).

3- In Figure 3, the authors show that CD62P and CD63 expression is increased in MPs from DV-infected platelets. Did DV infection change only the expression of surface molecules on MPs or also change the number of EVs released from infected platelets?

4- Platelets were recently shown to productively replicate the DV (Simon et al, Blood 2015). Since exosomes are in the same size range than DV, how can the authors differentiate exosomes from DV particles released from infected platelets? As shown in Figure 2 and 4, exosomes from infected platelets activated neutrophils depending mainly on CLEC-5 expression, which was very similar to

neutrophil activation by DV alone. The authors should consider the possibility that DV contamination in isolated exosomes are driving NET extrusion in this experiment.

5-Is the ability to induce NET a feature exclusive of CLEC-2-activated platelets? Are MPs from platelets activated by other agonists also able to induce NET?

6-The authors investigated the ability of EVs from infected platelets to increase endothelial cell permeability (Figure 5) or cytokine release (Figure 7) in vitro. However, they never describe whether they are using MPs or exosomes in this assay. Please clarify this issue.

7-Was vascular permeability in vivo (Figure 5D) and animal survival (Figure 7D) significantly reduced in stat1/clec5 double KO compared to stat1 KO mice? Was this difference still present after anti-TLR2 treatment?

Minor issues:

1-The discussion can be improved with information on other receptors for DV on platelets. In addition to CLEC-2, another C type lectin receptor, namely DC-SIGN, have been implicated in DV and HIV binding and internalization by platelets; and in DV-induced platelet activation (Simon et al, Blood 2015, Hottz et al, JTH 2013; Chaipan et al, J Vir 2006; Boukour et al, JTH 2006; reviewed in Hottz et al, Front Med 2018).

2-Increased levels of cell-free histones have been shown to correlate with disease severity in dengue. In addition, circulating cell-free histones contribute platelet activation in dengue patients (Trugilho et al, Plos Path 2017). The authors should discuss how platelet-induced NET extrusion makes a perfect explanation for increased levels of cell-free histones in dengue, and how histones can reciprocally increase platelet activation.

3-Line 109: 30 µg/well. Please correct to µg/mL.

4-Reference 17 does not show CLEC-2 involvement in HIV binding to platelets, as cited (line 250), please correct.

5-Figure 1 B: Error bars and statistics are missing.

Reviewer #4 (Remarks to the Author):

This is an interesting study showing that TLR2 and CLEC5a are PRRs for Dengue virus. The work shows that DV induces platelets to make exosomes and vesicles which induce neutrophils to make NETs. Both human and mouse systems are used. This is a well accomplished study. I have really only one issue with this entire study. The role of DNase-1.

1) It is not clear how adding DNase-1 which chops up DNA would reduce the amount of citrullinated- H3. This is surprising as DNase should only get rid of DNA.

2) Secondly the DNase reduced permeability and various other biology. How does this happen. Presumably it is not naked DNA that is the biological effector. Presumably proteases and various other molecules are released and so why is DNase so effective.

3) I would like to see how effective DNase-1 is in the mouse model for mortality and various other parameters.

Reviewer #1 (Remarks to the Author):

The manuscript entitled "CLEC2-CLEC5A/TLR2 axis is critical in dengue virus-induced inflammation and lethality" by Sung et al. proposed the novel axis of CLEC2-CLEC5A/TLR2. The authors proposed a novel CLEC2-CLEC5A/TLR2 axis as a critical signaling pathway for dengue virus-induced inflammation. The authors also mentioned that the simultaneous blockade of CLEC5A and TLR2 may reduce dengue virus-induced inflammation and lethality. Although the study is potentially important from the clinical point of view, this reviewer feels that the manuscript would be much benefitted if the authors provide additional clarifications to following concerns.

Comments

1) There is no abstract.

Ans: 'Abstract' is included in this revised version. We uploaded the abstract as a separate file in previous version, so it is missing in the text file of previous version.

2) No direct evidence is provided for the description "CLEC2-CLEC5A/TLR2 axis" throughout the manuscript.

Ans: Our results suggest that dengue virus (DV) activates CLEC2 in platelets to release extracellular vesicles, thereby enhance DV-induced NET formation and proinflammatory cytokine release via CLEC5A/TLR2. We agree with your comment, and change the original title to 'Extracellular vesicles from CLEC2-activated platelets enhance dengue virus-induced lethality via CLEC5A/TLR2'.

3) It would be better to improve the quality of Figure 4A.

Ans: We have repeated the experiments to improve the quality of Western blot. Please see Figure 4C in the revised manuscript.

4) Figure 4D is missing in the current manuscript.

Ans: We apologize the typo in previous version. The figure we refer is Figure 4D as shown in this revised version (line 177-179).

5) The authors may want to use Stat1^{-/-}TLR2^{-/-} mice to clarify the requirement of TLR2 in the infection experiment.

Ans: Because the stat1^{-/-}tlr2^{-/-} mice are not available, we injected anti-TLR2 mAb into stat1 KO mice to clarify the role of TLR2 in DV infection in vivo (Fig. 5D, Fig. 6, Fig. 7B~D).

In addition, we showed that tlr2^{-/-} neutrophils cannot produce NET in vitro (Fig. 2C). These observations support the argument that TLR2 plays less important role than CLEC5A in DV-induced lethality.

中央研究院
基因體研究中心
Genomics Research Center
ACADEMIA SINICA

Genomics Research Center,
Academia Sinica
128, Academia Road, Section 2
Nankang, Taipei 115, Taiwan R.O.C.
Tel/+886-2-27871245
Fax/+886-2-2789-8811
E-mail: slhsieh@gate.sinica.edu.tw

- 6) The each value in Figure 7B should be presented as actual concentration rather than fold change.
Ans: We follow the suggestion and present the data as actual concentration in this revised version (Figure 7B).

Reviewer #2 (Remarks to the Author):

In this paper the authors are interested in understanding how Dengue virus (DV) activates platelets and leukocytes and whether DV-induced leukocyte-platelet interaction contributes to disease severity. In this paper the authors show that DV activates a molecule called CLEC5A in neutrophils to induce NETosis which is further enhanced by the presence of platelets. It was already shown that CLEC5A can interact directly with the dengue virion and induce pro-inflammatory cytokine expression (Chen et al., Nature 2008). Moreover, anti-CLEC5A monoclonal antibodies inhibit DV-induced plasma leakage, and vital-organ haemorrhaging reducing the mortality of DV infection by about 50% in STAT1-deficient mice. The authors suggest that DV activates CLEC2 in platelets to release extracellular vesicles (EVs), which further enhance NET formation and pro-inflammatory cytokine production via activating CLEC5A and TLR2 in both macrophages and neutrophils. Blocking CLEC2-CLEC5A/TLR2 axis may be a novel strategy to reduce tissue damage of patients suffering from severe DV infection.

This is a beautiful story with clearly presented figures and the content is potentially highly relevant for our understanding of DV pathogenesis. While the data is compelling and demonstrates a role for a CLEC-2 platelet and CLEC5-neutrophil axis in NET formation, I have several questions about the mechanistic evidence that may need addressing.

Major concerns

1) The authors claim that DV-activated platelets require CLEC2 to generate EVs that in turn induce NETosis via CLEC5A. I am not convinced yet that alternative activation of platelets will not lead to similar NET formation. Could the authors show that CLEC2-independent activation of platelets (PMA/von Willebrand factor/PAR-1 agonists?) have similar effects?

Ans: We did not claim that alternative activation cannot of platelets cannot enhance NET formation in previous version. We only claim that DVs activate platelets via CLEC2 to enhance NET formation. Actually, our observation is in accord with reviewer's speculation. We did find that LPS- and thrombin-activated platelets also enhance NET formation via CLEC5A and TLR2 (as shown in the following figure). We point out this finding in the Discussion Session (line 281-285) to emphasize this phenomenon is not only limited to CLEC2-activated platelets.

Human neutrophils (8×10^5) were pretreated with isotype, or anti-CLEC5A mAb, or anti-TLR2 mAb or simultaneously for 1 hr at 37 °C. Autologous platelets (8×10^6) were co-cultured with neutrophils and stimulated with LPS (1 µg/ml) or thrombin (0.1 U/ml) for 3 hr at 37 °C.

2) The authors show that DV itself also activates NET formation (Figure 2A) which is increased by adding CLEC2-activated PLTs. It is unclear though how the authors separate DVs from the DV-activated PLTs and MVs. One may assume that the MVs and EXOs preparations will be contaminated with intact DVs. Can the authors rule out this possibility? How are the amount of material, PLTS, MVs and EXOs compared?

Ans:

(a) We examine the potential contamination of DV in DV-MVs and DV-EXOs by plaque assay using BHK21 cell line, and found the DV titer is approximately 150-200 PFU/ml. However, such low titer of DV is unable to induce NET formation and cytokine release. In addition, neutralization mAb against DV E protein is unable to inhibit DV-MVs and DV-EXOs-induced NET formation. These observations suggest that the enhanced NET formation is not due to the contaminated DVs. We also point out this issue in the Discussion Session (line 293-299) in the revised version.

b) It is hard to compare the amounts of EVs with platelets, thus we compared the effects of DV-activated platelets (8×10^6) and DV-EVs (1.5×10^8) from same amounts of platelets (8×10^6) for comparison in these experiments (Figure 3B). We also use the amount of DV-EXOs (1.5×10^8) and DV-MVs (1.5×10^8) to induce NET formation in Figure 4D&4E.

3) While CLEC2 mAb and CLEC2^{-/-} platelets lose their ability to cause NET formation, it remains unclear what the critical molecule(s) on the platelets and or EVs is/are that support (CLEC5-dependent) NET formation. Can the authors identify critical molecules on the PLTs/MVs or EXOs that activate the NET formation? I am asking because one may hypothesize that CLEC2 activation of PLTs may induce selected molecules for secretion via EVs.

*Ans: We performed mass spectrometry to analyze the components of MVs and EXOs from unstimulated platelets, aggretin-activated platelets, and DV-activated platelets. The following proteins are upregulated dramatically in both aggretin- and DV-activated platelets. Thus, these molecules are the potential 'danger signals' from CLEC2-activated platelets to induce NET formation. We provide this information in **Table S1** and point out this issue in Discussion Session of this revised version (line 301-324).*

	Protein name	Gene name	Increase/Decrease	Significantly different compared to mock (Score)
MVs	Tubulin beta-1 chain	TUBB1	↑	28.166
	Guanine nucleotide-binding protein G(I)/G(S)/G(O) subunit gamma-3	GNG3	↑	6.0121
	Tribbles homolog 1	TRIB1	↑	5.8874
EXOs	Vinculin	VCL	↑	21.874
	Coagulation factor XIII A chain	F13A1	↑	18.076
	Calnexin	CANX	↑	11.591

Minor

1) What is the concentration of isotype control in Fig 2A?

Ans: The concentration of isotype is 300 µg/ml as shown in the legend of Figure 2A (line 611).

2) The western blots of the MVs and EXs are not extremely convincing, in particular CD63 bands are suboptimal what is the predicted size maybe use non-reducing conditions. In addition, other EV

protein markers such as HSP70 or CD81 and TSG101 or ALIX should be shown.

Ans: We performed the western blotting experiments using non-reducing condition as reviewer's suggestion, and detect the presence of HSP70 and CD81, in addition to CD63, CD9 and CD41, in human platelets-derived extracellular vesicles (Figure. 4C). This result is in accord with what we observed in mass spectrometry assay. However, we did not detect TSG101 and ALIX in mass spectrometry assay, so we did not probe these two proteins again in the Western blot analysis. Because the platelets are from freshly isolated human blood, and the amount of EXOs and MVs are very limited after sequential ultracentrifugation, thus unable to get strong signals in previous version.

3) Finally, it would be better if the authors to show Electron Microscopy images of the PLT, MV and EXO preparations to get an idea of their composition/purity.

Ans: We did observe MV and EXO under a transmission electron microscope. We provide this information in Figure S4 and describe what we observe in the Discussion Session (line 314-327).

Reviewer #3 (Remarks to the Author):

Sung and colleagues analyzed the involvement of CLEC-2 and CLEC-5 in DENV-induced platelet activation and platelet-mediated NET extrusion. Similar to platelets, platelet-derived extracellular vesicles were also able to increase NET extrusion. The participation of platelet CLEC-2 and neutrophil CLEC-5 and TLR-2 in NET extrusion in vitro is very convincing. The participation of CLEC-5 and TLR-2 in NET extrusion and increased vascular permeability is also very convincing in vivo. However, there is no evidence for the participation of platelets or platelet-CLEC-2 in the release of NET, increased vascular permeability or mortality in vivo. Even though the data presented here are of interest, some concerns must be addressed.

Major issues:

1- The authors mention platelet-specific CLEC-2 KO mice (CLEC2^{flox}/PF4^{cre}) in material and methods (line 337). Were platelet-specific CLEC-2 KO mice protected from DV infection? Were they susceptible to platelet activation, NET extrusion in vivo and increased vascular permeability? In addition, platelet depletion experiments would be useful to clarify the importance of platelets to NET extrusion and vascular damage in vivo.)

Ans:

a) Even though DVs can stimulate platelets isolated from WT and CLEC2 KO mice in vitro, WT and CLEC-2 KO mice are not susceptible to DV infection in vivo. Thus, we are unable to address the role of CLEC2 in DV infection using platelet-specific CLEC-2 KO mice. Therefore, we inject anti-CLEC2 mAb into Stat1 KO mice to address this question (Figure 5D, Figure 7, Figure B~D).

b) We deplete platelet in *stat1*^{-/-} and *stat1*^{-/-}*clec5A*^{-/-} mice, and found all the mice died within 24 hours. This observation is in accord with previous observation that the thrombocytopenia is associated with disease susceptibility (DOI: <http://dx.doi.org/10.18203/2320-6012.ijrms20171543>) (<https://www.msjonline.org/index.php/ijrms/article/view/2991>).

c) It is difficult to get the platelet-specific CLEC-2 KO mice in Stat1 background. Therefore, we test the role of CLEC2 in virus infection using H5N1 influenza virus, which can infect platelet-specific CLEC-2 KO mice in B6 background (Dengue virus cannot infect wild type B6 background). We found the survival rate of *clec2*^{-/-} mice is higher than wild type mice (85.7% vs. 62.5%, left panel). Compared to WT platelet, influenza virus activated *clec2*^{-/-} platelets did not enhance NET formation (right panel).

2-Identification of extracellular vesicles (EVs) as microparticles (MPs) or exosomes is a crucial

@g

aspect of the author's conclusions. Importantly, the authors do not present any data on characterization of platelet-derived MPs and exosomes (size, number, surface markers, ultrastructure...).

Ans:

We did characterize platelet-derived EXOs and MVs and provide this information in the revised version. The information from NTA assay is shown in Figure 4A&4B, and marker expression is shown in Figure 4C. We also observe MVs under a transmission electron microscope (Figure S4) and describe our finding in the Discussion Session (line 314-327). We also performed mass spectrometry to analyze the components of EXOs and MVs in Table S1 and Discussion Session of this revised version (line 301-324).

3-In Figure 3, the authors show that CD62P and CD63 expression is increased in MPs from DV-infected platelets. Did DV infection change only the expression of surface molecules on MPs or also change the number of EVs released from infected platelets?

Ans: DV not only upregulate the expression of CD62P and CD63, but also increase the number of EVs. This information is shown in Figure 1A (upper right panel) in the revised version.

4-Platelets were recently shown to productively replicate the DV (Simon et al, Blood 2015). Since exosomes are in the same size range than DV, how can the authors differentiate exosomes from DV particles released from infected platelets as shown in Figure 2 and 4, exosomes from infected platelets activated neutrophils depending mainly on CLEC-5A expression, which was very similar to neutrophil activation by DV alone. The authors should consider the possibility that DV contamination in isolated exosomes are driving NET extrusion in this experiment.

Ans:

We examine the potential contamination of DV in DV-MVs and DV-EXOs by plaque assay using BHK21 cell line, and found the DV titer is approximately 150-200 PFU/ml. However, such low titer of DV is unable to induce NET formation and cytokine release. In addition, neutralization mAb against DV E protein is unable to inhibit DV-MVs and DV-EXOs-induced NET formation. These observations suggest that the enhanced NET formation is not due to the contaminated DVs. We also point out this issue in the Discussion Session (line 293-299) in the revised version.

5-Is the ability to induce NET a feature exclusive of CLEC-2-activated platelets? Are MPs from platelets activated by other agonists also able to induce NET?

Ans:

We did find that LPS- and thrombin-activated platelets also enhance NET formation via CLEC5A and TLR2 (as shown in the following figure). We point out this finding in the Discussion Session (line 281-285) to emphasize this phenomenon is not only limited to CLEC2-activated platelets. The data are shown in the following:

Human neutrophils (8×10^5) were pretreated with isotype, or anti-CLEC5A mAb, or anti-TLR2 mAb or simultaneously for 1 hr at 37 °C. EVs from LPS (1 µg/ml)- or thrombin (TH, 0.1 U/ml)-activated platelets (8×10^6) were co-cultured with neutrophils for 3 hr at 37 °C.

6-The authors investigated the ability of EVs from infected platelets to increase endothelial cell permeability (Figure 5) or cytokine release (Figure 7) in vitro. However, they never describe whether they are using MPs or exosomes in this assay. Please clarify this issue.

Ans:

We use DV-activated extracellular vesicles (including both EXOs and MVs) to perform the experiments for Figure 5 and Figure 7. We revise Figure 5 (line 655-658) and Figure 7 (line 684-686) legends to make this point clear.

7-Was vascular permeability in vivo (Figure 5D) and animal survival (Figure 7D) significantly reduced in stat1/clec5 double KO compared to stat1 KO mice? Was this difference still present after anti-TLR2 treatment?

Ans:

a. Yes, *stat1*^{-/-}*clec5a*^{-/-} significantly produced lower NET formation than *stat1*^{-/-} mice

b. We had compared the Evans blue level between the *stat1*^{-/-} and *stat1*^{-/-}*clec5a*^{-/-} in TLR2 mAb-treated groups.

Minor issues:

1-The discussion can be improved with information on other receptors for DV on platelets. In addition to CLEC-2, another C type lectin receptor, namely DC-SIGN, have been implicated in DV and HIV binding and internalization by platelets; and in DV-induced platelet activation (Simon et al, Blood 2015, Hottz et al, JTH 2013; Chaipan et al, J Vir 2006; Boukour et al, JTH 2006; reviewed in Hottz et al, Front Med 2018).

Ans: We discuss the role of other C-type lectins in viral infection accordingly, and include the above references in the Discussion Session (Line 338-346).

2-Increased levels of cell-free histones have been shown to correlate with disease severity in dengue. In addition, circulating cell-free histones contribute platelet activation in dengue patients (Trugilho et al, Plos Path 2017). The authors should discuss how platelet-induced NET extrusion makes a perfect explanation for increased levels of cell-free histones in dengue, and how histones can reciprocally increase platelet activation.

Ans: Yes, we agree with the comment and suggestion. We point out that NET formation may contribute to the increased histone amount in serum of DV-infected patients, and discuss the potential role of histone to enhance dengue infection in the Discussion Session (line 346-349).

3-Line 109: 30 µg/well. Please correct to µg/mL.

Ans: We used to put fixed amount (30 ug) of antibody per well (100 ul), so we convert from 30 µg/well into 300 µg/ml in this revised version (line 125). We change the fixed amount into concentration as the reviewer's suggestion in all the revised manuscript, including 'figure legend'(line 611).

4-Reference 17 does not show CLEC-2 involvement in HIV binding to platelets, as cited (line 250), please correct.

Ans: We are sorry for the mistake. We correct the citation number to #22 in this revised version (line

272).

5-Figure 1 B: Error bars and statistics are missing.

Ans: Error bars and statistics are included in Fig. 1B in the revised version.

Reviewer #4 (Remarks to the Author):

This is an interesting study showing that TLR2 and CLEC5a are PRRs for Dengue virus. The work shows that DV induces platelets to make exosomes and vesicles which induce neutrophils to make NETs. Both human and mouse systems are used. This is a well accomplished study. I have really only one issue with this entire study. The role of DNase I.

1) It is not clear how adding DNase I which chops up DNA would reduce the amount of citrullinated-H3. This is surprising as DNase I should only get rid of DNA.

Ans: Because citrullinated histone is associated with DNA, removal of DNA by DNase I also release citrullinated histone. The effect of DNase I to reduce citrullinated histone is also reported by other group. Following is the figure 1 from Dr. Ting's group (Scientific reports 8: 17788, 2018)

Figure 1. Identification of NETs *in vivo* by immunofluorescence staining. (A) Rats were subjected to intestinal I/R injury to induce neutrophil infiltration (MPO positive, green, n = 4 per group). Scale bar: 20 μm. (B) NETs were detected after 1 h ischemia and 2 h reperfusion (Cit-H3-positive, green, n = 4 per group). Scale bar: 20 μm.

2) Secondly the DNase1 reduced permeability and various other biology. How does this happen. Presumably it is not naked DNA that is the biological effector. Presumably proteases and various other molecules are released and so why is DNase I so effective.

Ans: We observed that DNase I reduced permeability change in vitro (Figure 5A) and in vivo (Figure. 5C). NETs have been shown to be cytotoxic to endothelium¹ and increase permeability change² (see reference 1&2 in the following page). Even though it has been speculated that factors released from neutrophil also contributes to these phenomena, it does not exclude the role of DNA, which associated with MPO, elastase, and other factors, in NET-mediated cytotoxicity and permeability change. Thus, our observation suggests that the integrity of NET structure, not the free form individual component, is required for NET-mediated cytotoxicity and permeability change.

3) I would like to see how effective DNase-1 is in the mouse model for mortality and various other parameters.

Ans: We did find that DNase I reduce mice permeability change (Figure 5C) and mortality rate (Figure S3) in DV-infected Stat1 KO mice. We point out this issue in Discussion Session (line 262-268).

Reference:

- 1 Gupta, A. K. *et al.* Activated endothelial cells induce neutrophil extracellular traps and are susceptible to NETosis-mediated cell death. *FEBS Lett* **584**, 3193-3197, doi:10.1016/j.febslet.2010.06.006 (2010).
- 2 Cadrillier, A. *et al.* Platelets induce neutrophil extracellular traps in transfusion-related acute lung injury. *J Clin Invest* **122**, 2661-2671, doi:10.1172/JCI61303 (2012).

REVIEWERS' COMMENTS:

Reviewer #1 (Remarks to the Author):

The authors have adequately addressed some of the concerns raised by this reviewer.

I think it is necessary for the authors to perform a binding assay of DV-CLEC2 and EVs-CLEC5A using some methods such as fusion-Fc protein binding assays. Indeed, Watson, et al. checked for CLEC5A and CLEC2 (2011 JBC). Only CLEC5A showed a significant binding to DV (all type) but not CLEC2. The authors should clarify these discrepancies.

The authors may want to correct line 178. anti-CLEC5A instead of ani-CLEC5A.
The authors may want to correct the line 317 sentence.

Reviewer #2 (Remarks to the Author):

The authors responded adequately to my questions.

I do however urge the authors to expand a bit in the discussion on the putative ligands on the MVs/exosomes for TLR2 and CLEC5a. While the authors propose based upon a proteomic EV analysis that proteins maybe involved, other, alternative ligands/biomolecules are not mentioned or considered. For example EV-enriched glycans may have a role as reviewed by Williams et al., JEV 2018.

Reviewer #3 (Remarks to the Author):

The revised version was much improved and the concerns raised were adequately addressed. No further concerns.

Reviewer #1 (Remarks to the Author):

The authors have adequately addressed some of the concerns raised by this reviewer.

I think it is necessary for the authors to perform a binding assay of DV-CLEC2 and EVs-CLEC5A using some methods such as fusion-Fc protein binding assays.

1) Watson, et al. checked for CLEC5A and CLEC2 (2011 JBC). Only CLEC5A showed a significant binding to DV (all type) but not CLEC2. The authors should clarify these discrepancies.

Ans: We address this issue and include the related references in Discussion Session Line 311 to 325.

2) The authors may want to correct line 178. anti-CLEC5A instead of ani-CLEC5A.

The authors may want to correct the line 317 sentence.

Ans: We correct this typo (line 179 and line 357).

Reviewer #2 (Remarks to the Author):

The authors responded adequately to my questions.

I do however urge the authors to expand a bit in the discussion on the putative ligands on the MVs/exosomes for TLR2 and CLEC5a. While the authors propose based upon a proteomic EV analysis that proteins maybe involved, other, alternative ligands/biomolecules are not mentioned or considered. For example, EV-enriched glycans may have a role as reviewed by Williams et al., JEV 2018.

Ans: We address this issue and include the reference in Discussion Session Line 347 to 352.

Reviewer #3 (Remarks to the Author):

The revised version was much improved and the concerns raised were adequately addressed. No further concerns.

Ans: Thanks.

Reviewer #4:

We ask that you incorporate in the main text a discussion regarding the role of DNase and the potential for other factors to be involved in the observed effects. Please ensure that these limitations, including a modest effect on survival, are stated clearly in the manuscript text and incorporate all relevant citations that support the role of DNase. Please provide a detailed point-by-point response to the reviewers'

and our requests while highlighting all changes made in the main manuscript file.

Ans: A paragraph to discuss the effect of DNase I in mice survival is shown in Discussion Session (Line 268 to 286).